# Calving front monitoring at a subseasonal resolution: a deep learning application for Greenland glaciers

**Erik Loebel[1,⚲], Mirko Scheinert[1], Martin Horwath[1], Angelika Humbert[2,3], Julia Sohn[2,4], Konrad Heidler[5], Charlotte Liebezeit[1], and Xiao Xiang Zhu[5]**

[1]Institut für Planetare Geodäsie, Technische Universität Dresden, Dresden, Germany
[2]Sektion Glaziologie, Alfred-Wegener-Institut, Helmholtz-Zentrum für Polar- und Meeresforschung, Bremerhaven, Germany
[3]Fachbereich Geowissenschaften, Universität Bremen, Bremen, Germany
[4]Professur für Engineering, IU Internationale Hochschule, Erfurt, Germany
[5]Chair of Data Science in Earth Observation, Technische Universität München, München, Germany
[⚲]Invited contribution by Erik Loebel, recipient of the EGU Cryospheric Sciences Virtual Outstanding Student and PhD candidate Presentation Award 2021.

**Correspondence:** Erik Loebel (erik.loebel@tu-dresden.de)

**Abstract.** ⬛TSI The mass balance of the Greenland Ice Sheet is strongly influenced by the dynamics of its outlet glaciers. Therefore, it is of paramount importance to accurately and continuously monitor these glaciers, especially the variation in their frontal positions. A temporally comprehensive parameterization of glacier calving is essential for understanding dynamic changes and constraining ice sheet modeling. However, many current calving front records are limited in terms of temporal resolution as they rely on manual delineation, which is laborious and not appropriate considering the increasing amount of satellite imagery available. In this contribution, we address this problem by applying an automated method to extract calving fronts from optical satellite imagery. The core of this workflow builds on recent advances in the field of deep learning while taking full advantage of multispectral input information. The performance of the method is evaluated using three independent test datasets. For the three datasets, we calculate mean delineation errors of 61.2, 73.7, and 73.5 m, respectively. Eventually, we apply the technique to Landsat-8 imagery. We generate 9243 calving front positions across 23 outlet glaciers in Greenland for the period 2013–2021. Resulting time series not only resolve long-term and seasonal signals but also resolve subseasonal patterns. We discuss the implications for glaciological studies and present a first application for analyzing the effect of bedrock topography on calving front variations. Our method and derived results represent an important step towards the development of intelligent processing strategies for glacier monitoring, opening up new possibilities for studying and modeling the dynamics of Greenland's outlet glaciers.

## 1 Introduction

Over the past 2 decades, the Greenland Ice Sheet has been a major contributor to sea level rise (Horwath et al., 2022). Models suggest that this imbalance will continue with a warming climate (Goelzer et al., 2020; Edwards et al., 2021; Rückamp et al., 2020). About half of the ice mass loss is due to increased meltwater runoff, while the other half is due to changes in ice discharge to the ocean, which are related to changes in the ice flow dynamics of outlet glaciers (Otosaka et al., 2023). Several mechanisms act as controls and indicators for dynamic glacier changes. In particular, calving and calving front variations have been identified as crucial parameters for investigating the physical mechanisms of Greenland's outlet glaciers (Joughin et al., 2008a; Moon and Joughin, 2008; Benn et al., 2017; Trevers et al., 2019; Cook et al., 2021; Melton et al., 2022). In addition, recent studies have shown that calving front retreat is associated with increased ice discharge (King et al., 2018;

Mouginot et al., 2019; King et al., 2020). An accurate representation of calving front behavior is therefore an important requirement for constraining ice sheet modeling and improving simulations of future mass loss and sea level contribution (Vieli and Nick, 2011; Bondizo et al., 2017; Morlighem et al., 2017, 2019; Greene et al., 2024). Overall, temporally and spatially comprehensive data products that consider calving front variation are essential for a better understanding and modeling of marine-terminating glaciers.

The steady increase in the quality and availability of satellite imagery provides new opportunities for a continuous and accurate monitoring of glacier calving front positions. Nevertheless, current data records mostly rely on manual delineation (Schild and Hamilton, 2013; Joughin et al., 2015; ENVEO, 2017; Andersen et al., 2019; King et al., 2020; Goliber et al., 2022; Black and Joughin, 2023). This is a laborious, time-consuming, and thus ineffective process given the ever-increasing volume of data. Therefore, such calving front products may not always fully exploit the temporal information offered by satellite observations, which may be a limiting factor in seasonal analyses and associated modeling efforts. In response to the need for scalable processing strategies, several empirical feature extraction algorithms have been introduced over the last decades, all aiming to provide robust automated calving front extraction (Sohn and Jezek, 1999; Liu and Jezek, 2004; Seale et al., 2011; Rosenau, 2014; Krieger and Floricioiu, 2017; Liu et al., 2021). However, most of these methods have not been tested for spatial transferability and large-scale applications, or they require case-specific modifications. With the advent of deep learning and big-data methods in remote sensing, new opportunities have emerged for solving complex image-processing tasks (Zhu et al., 2017). In recent years, a number of case studies have used deep artificial neural networks (ANNs) to extract calving front positions. Both optical (Mohajerani et al., 2019) and synthetic aperture radar (SAR) (Zhang et al., 2019; Baumhoer et al., 2019) sensors have been used CE1. Based on these case studies, numerous studies have advanced the ANN architecture (Heidler et al., 2021; Marochov et al., 2021; Periyasamy et al., 2022; Davari et al., 2022b, a; Heidler et al., 2023; Herrmann et al., 2023; Wu et al., 2023), assessed potential input information (Loebel et al., 2022), and pursued multisensor capabilities (Zhang et al., 2021). In addition, dedicated data products have been developed for training and validation (Goliber et al., 2022) as well as for benchmarking (Gourmelon et al., 2022) ANN applications. The results from Cheng et al. (2021) and Zhang et al. (2023), specifically the Calving Front Machine (CALFIN) and AutoTerm repositories, are currently the only two automatically generated datasets that provide a Greenland-wide scope of calving front locations. Building on these achievements, this paper discusses the application and extensive validation of a specially tailored deep learning method for automating calving front extraction using Landsat-8 optical imagery. We provide a data product for 23 outlet glaciers in Greenland for the period 2013–2021. We compare this data product with the automatically delineated CALFIN and AutoTerm repositories as well as with the manually delineated TermPicks and Black and Joughin (2023) repositories. By exploiting the full multispectral sensor information, our method is able to extract a significant number of calving fronts that could not be extracted using the other automation methods. By achieving this method of robust and scalable calving front extraction, we meet the glaciology community's requirement for a comprehensive parameterization of glacier calving in Greenland and make important steps towards establishing artificial-intelligence-based processing strategies for glacier-monitoring tasks. Overall, we provide the wider cryosphere community with a methodology, a data product and its implementation, a comparison with existing products, and a discussion of the glaciological implications.

Section 2 introduces the data and the applied deep learning method for automated calving front extraction. Section 3 provides an assessment of the accuracy of our method and its spatial transferability. As part of the discussion in Sect. 4, we present our data product, the derived time series, and a comparison with existing data repositories. apply our results to analyze the interaction between calving front change and bedrock topography. Finally, in Sect. 6, we draw conclusions and provide an outlook.

## 2 Data and methods

The presented processing system extracts glacier calving front shapefiles from multispectral Landsat-8 imagery. In this process, we use satellite imagery as reference data and apply a specialized ANN. This involves a series of processing steps and configurations, which are explored in the following section.

### 2.1 Data source

Our processing system is based on optical Landsat-8 imagery. We use the orthorectified and radiometrically calibrated Level-1 data products provided by the United States Geological Survey (U.S. Geological Survey, 2023). Carrying two scientific instruments, the Operational Land Imager (OLI) and the Thermal Infrared Sensor (TIRS), the Landsat-8 satellite provides a particularly wide multispectral coverage. The 11 spectral bands comprise data from visible, near-infrared, shortwave infrared, and thermal infrared wavelengths (from 0.435 TS2 to 1.384 µm). With the exception of the panchromatic band and the two thermal bands, which have a spatial resolution of 15 and 100 m, respectively, all other bands have a resolution of 30 m. All available bands (except band 8 and band 9) are used as input for our ANN. Band 8, which has a 15 m resolution, is excluded due to its high computational cost. Band 9 is outside an atmospheric

window and is therefore intended for atmospheric observations. Integrating multispectral bands generally leads to more accurate predictions than using conventional single-band inputs alone, as demonstrated by Loebel et al. (2022). This is especially true for difficult illumination and ice mélange conditions.

## 2.2 Reference dataset

We use manually delineated calving front locations as reference data. For model training, we use 698 calving front positions across 19 glaciers in Greenland for the period 2013–2019. These glaciers have been selected for their broad spatial distribution and diverse morphology as well as for their differing calving and ocean conditions. A spatial overview of all glaciers in Greenland analyzed in this study is given in Fig. 1. As the performance of ANN methods highly depends on training data, we pay special attention to covering a diverse range of morphological features, termini with heavy crevassing, and differing calving and ice mélange conditions, as well as varying illumination and cloud situations. To test the model, we apply three different testing sets. The Technische Universität Dresden (TUD) testing dataset includes four additional glaciers in Greenland, the Boydell and Drygalski glaciers in the Antarctic Peninsula, the Storbreen glacier in Svalbard, and the Upsala Glacier in Patagonia. In total, the TUD testing set contains 200 calving front positions across 27 glaciers for the period 2020–2021. In addition to our own testing dataset, we use manually delineated calving fronts from the ESA's Climate Change Initiative (ESA-CCI) (ENVEO, 2017) and the CALFIN product (Cheng et al., 2021). Here, we use all available calving front positions for our selected Greenland-based glaciers that have a corresponding Landsat-8 scene with less than 24 h of time difference. This results in a further 100 manually delineated calving front positions for the ESA-CCI and a further 110 for the CALFIN testing datasets.

## 2.3 Delineating calving fronts using deep learning

For automated calving front extraction, we apply a modified version of the approach published by Loebel et al. (2022). The main difference is that we only use multispectral information and no textural or topographic features. This reduces the input from 17 to 9 layers. Additionally, we have expanded the reference dataset by 170 entries. These new calving front traces focus specifically on cloudy, low illumination and scene border conditions, thereby enhancing the method in this regard. Figure 2 presents a broad overview of the processing workflow.

### 2.3.1 Preprocessing

Using satellite data as input for the ANN requires preprocessing. Specifically, we create stacked raster subsets from the multispectral satellite bands and the manually delineated

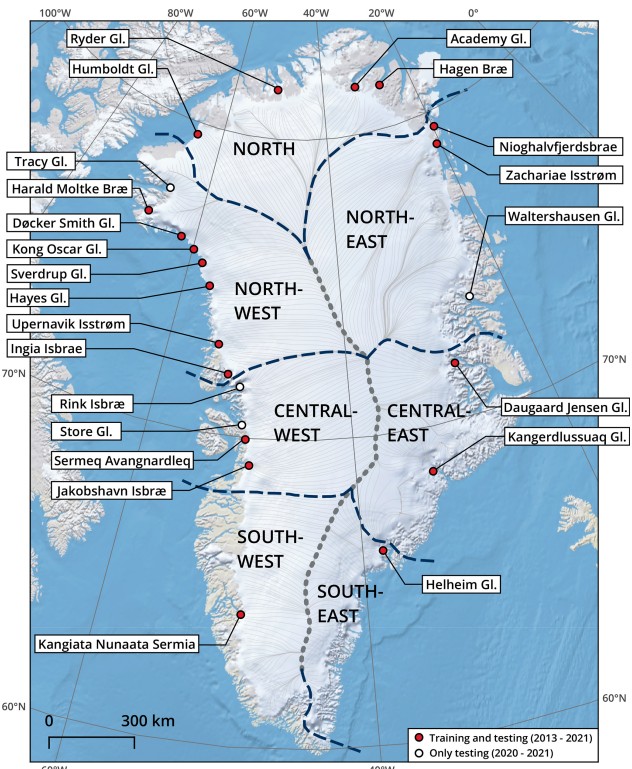

**Figure 1.** CE2 Overview map of the 23 Greenland-based glaciers used in the TUD reference dataset. Glaciers marked with a red dot are used for training and testing. White dots indicate the glaciers that are only used for model testing. The Boydell Glacier (Antarctica), Drygalski Glacier (Antarctica), Storbreen glacier (Svalbard), and Upsala Glacier (Patagonia) are not on this map and are only used for model testing. The base map is from the "QGreenland" package (Moon et al., 2022).

calving front locations. These subsets have dimensions of $512\,\text{px} \times 512\,\text{px}$ and a unified $30\,\text{m}$ ground sampling distance, and they are centered on the calving front of the respective glacier. The $30\,\text{m}$ ground sampling distance, and thus the exclusion of band 8, is a compromise between the spatial context provided within a single subset, the computational effort, and the resolution of the calving front predictions. For each multispectral band, we apply an image enhancement technique in the form of a cumulative count cut, clipping the data between the 0.1 and 98 percentiles to counteract overexposure in our satellite imagery. Additionally, all satellite bands are then normalized to a range between 0 and 1 using 8-bit quantization. Corresponding manually delineated calving front positions, given as either line strings or polygon shapefiles, are processed into binary raster masks that segment land and glaciers from the ocean. Altogether, one stacked raster subset includes nine satellite bands and a matching ground truth mask.

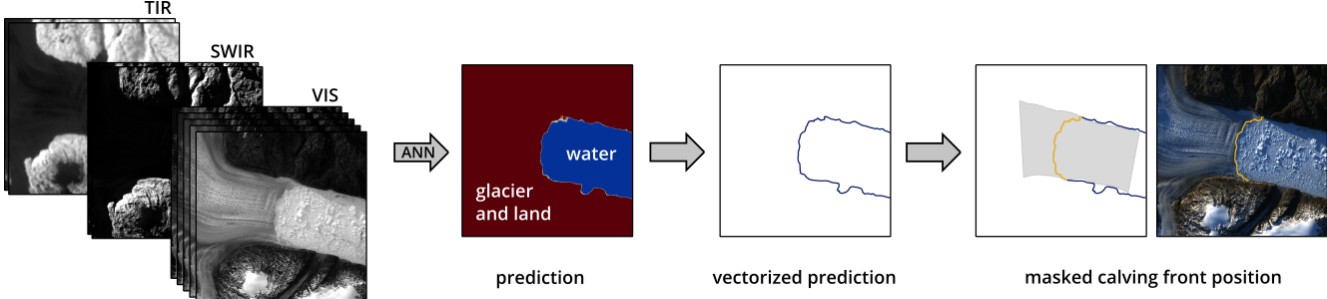

**Figure 2.** High-level overview of the applied workflow for automated calving front extraction. Using multispectral satellite imagery at visible (VIS), shortwave infrared (SWIR), and thermal infrared (TIR) wavelengths, the ANN performs pixel-wise semantic image segmentation. The final calving front position is obtained after vectorizing and masking the ANN prediction. Landsat-8 imagery is courtesy of the U.S. Geological Survey.

### 2.3.2 Semantic image segmentation

To extract the calving front location from the input images, we apply a convolutional neural network that performs pixel-wise semantic image segmentation, separating a glacier–land class from a water class. Specifically, we use a U-Net architecture introduced by Ronneberger et al. (2015). This architecture consists of a contracting path, similar to a typical convolutional network, where spatial resolution is reduced and feature information is increased, followed by an expanding path where feature and spatial information are combined. The receptive field of a U-Net is defined by the number of contracting and expanding blocks. As calving front extraction requires adequate spatial context (Heidler et al., 2021), in this study, we enhance the U-Net by two additional resolution levels, i.e., from the fourth to the sixth level. Our model is fitted using the preprocessed training data. Before initializing the model training, we select every fifth image from the training dataset for internal validation. The remaining training data are augmented 8-fold through rotation and flipping. Finally, the resulting 6208 raster subsets are used to fit the model. For this, we use randomized batches (each containing eight samples) and apply the Adam optimization algorithm (Kingma and Ba, 2014) to a binary cross-entropy loss function for a total of 200 epochs. Final model weights are selected based on the classification accuracy of the internal-validation dataset. ANN processing is implemented using the TensorFlow 2.4 library (Abadi et al., 2015). Model training is carried out using an IBM POWER9 node and an NVIDIA V100 GPU with a high memory bandwidth of 32 GB. The training of one model requires about 12 h, with a main memory utilization of 80 GB and an average GPU power consumption of 265 W.

### 2.3.3 Postprocessing

As output from the ANN, we derive a floating-point-number probability mask, where each image pixel is assigned a probability between 0 (water) and 1 (glacier and land). During postprocessing, we vectorize this probability mask using the Geospatial Data Abstraction Library (GDAL) contour algorithm (GDAL/OGR contributors, 2020) with a threshold of 0.5 and separate the longest feature. Eventually, we extract the glacier's calving front by intersecting the vectorized coastline trajectory with a static mask. This mask is created manually for each glacier and specifies a corridor of possible calving front locations. Calving fronts exceeding the 512 px × 512 px window are split into multiple independent predictions, which are then averaged in the overlapping area before vectorization. By applying this strategy, which is motivated by Baumhoer et al. (2019), the Zachariae Isstrøm, Nioghalvfjerdsbrae, and Humboldt Glacier are split into two, three, and seven separate overlapping predictions, respectively.

## 3 Accuracy assessment

Our own TUD testing set contains 200 labeled images from the years 2020 and 2021. We emphasize that the images are temporally separated from those in the training datasets. To ensure the spatial transferability of our method, this test dataset includes imagery of four additional glaciers in Greenland, two glaciers in the Antarctic Peninsula, one glacier in Svalbard, and one glacier in Patagonia. In addition to our own testing dataset, we apply a further 100 manually picked calving fronts provided by the ESA Greenland Ice Sheet CCI project (ENVEO, 2017) and a further 110 provided by the CALFIN product (Cheng et al., 2021).

### 3.1 Error estimation

The distance between the predicted and manually delineated calving fronts is taken as the main error measure in the model testing. We calculate the average minimal-distance error by averaging the minimal distances between the predicted front trajectory and the manual delineation calculated every 30 m. Our definition of the average minimal-distance error is com-

parable to the estimates used by Cheng et al. (2021) and Zhang et al. (2023). Figure 3 illustrates some test results for diverse testing images from the three test sets. Along with the manually picked calving front (dashed black) and the ANN-delineated calving front (orange), the average distance between them is indicated. The ANN model reliably delineates calving front locations in a wide range of ocean, sea ice, and ice mélange situations. Furthermore, the model is also able to handle images affected by challenging cloud (Fig. 3d, j) and illumination (Fig. 3c) conditions, as well as those affected by calving fronts near the edge of a satellite scene (Fig. 3e). Test images showing large errors are associated with delineation subjectivity (Fig. 3f, h, and i) or even human error (Fig. 3k).

Since the ANN training is stochastic, every fitted model performs slightly differently when using our testing data. To ensure statistical stability for a broader numerical assessment, we train and test 50 models using the same reference data and model parameters. In order to assess the distance error, we report both the mean and median across the scenes in the test dataset. The test results for these 50 models are shown in Fig. 4. While the mean distance error is sensitive to outliers, the median distance error informs us about systematic model overfitting and general scene-by-scene performance. Since each of the three testing datasets originated from its own independent imagery, resulting error estimates are not directly comparable. Nevertheless, we suspect that the lower distance error yielded for the TUD testing set is due to the fact that it was generated by the same people who inferred the training data for these models. Overall, the means of the average minimal-distance errors are comparable to the results from Cheng et al. (2021) and (Zhang et al., 2023), who estimated errors of $86.7 \pm 1.4$ and 79 m, respectively. Table 1 gives the corresponding statistics.

In addition to the average minimal-distance estimates, we also calculate the Hausdorff distance (Huttenlocher et al., 1993). The Hausdorff distance only considers the greatest distance of all minimum distances along the two trajectories. As longer fronts are more likely to include misclassified parts, this measure tends to be larger for longer fronts. Goliber et al. (2022) applied a median Hausdorff distance to duplicated delineations from different authors in order to estimate the accuracy level of manual digitization. Depending on the paired authors, this manual-delineation error varies between 59 and 7350 m, with an average of 107 m. The median Hausdorff distances calculated for our test data are therefore within the range of the manual-delineation errors but are slightly larger than the overall author-to-author error of 107 m calculated by Goliber et al. (2022). Altogether, the quality of calving fronts delineated by our ANN model is comparable to that of manually delineated calving fronts.

## 3.2 Spatial transferability

In addition to the accuracy assessment over the entire test dataset, we evaluate the degree of model generalization and, hence, the spatial transferability of our method. Out of our 200 test scenes, 61 are from glaciers that are not included in the training data. For these 61 test scenes over our 50 trained models, we calculate a mean (median) average minimal-distance error of $71.3 \pm 19.4$ m ($24.6 \pm 2.1$ m). This test error is larger than the error over the entire test set, which is $61.2 \pm 7.5$ m. It is thus also larger than the error over the 139 test scenes from glaciers that are part of the training set, which is $56.0 \pm 5.3$ m (median: $30.3 \pm 1.7$ m). Notably, we observe not only a larger test error but also a higher standard deviation between the models. This is due to a lower success rate and the resulting high error for individual predictions in cases where the ANN failed to locate the calving front.

Figure 5 presents the test results for four example scenes. The depicted glaciers are outside the training dataset. The calving fronts of the Tracy Glacier (Fig. 5a), Upsala Glacier (Fig. 5c), and Drygalski Glacier (Fig. 5d) are reliably extracted with low distances to the manually delineated reference and low deviation among all trained models. The accuracy is comparable to that of glaciers within the training dataset. In contrast, the extractions for the Storbreen glacier (left-hand side of Fig. 5b) exhibit a large error and high deviation among the trained models. The calving front is not delineated reliably. This could be due to a combination of difficult lighting conditions and snow-covered sea ice, which may not be adequately represented in the training data. Interestingly, the calving front of the neighboring Hornbreen glacier (bottom-right corner of Fig. 5b) is extracted accurately across all models.

Among the 61 test images from glaciers outside the training dataset, 57 have an average minimal-distance error below 100 m (93 %), compared to 178 out of 200 images from the entire test dataset (89 %) and 121 out of 139 test images from glaciers included in the training set (87 %). Overall, this assessment confirms the spatial transferability of our processing system. However, the accuracy is lower compared to that of the extraction from glaciers included in the training data. Similar findings have been reported by previous studies (Baumhoer et al., 2019; Cheng et al., 2021; Zhang et al., 2023).

## 4 Results

### 4.1 Data product for Greenland from 2013 to 2021

After training and testing the ANN model, we apply our processing to Landsat-8 imagery in order to generate temporally dense calving front time series for 23 outlet glaciers in Greenland. In doing so, we download Landsat-8 imagery acquired between March 2013 and December 2021. Images with cloud cover greater than 20 % and all Systematic Terrain Correction (L1GT) scenes are manually checked before being downloaded. Depending on the glacier, 51 % (Ingia Isbræ) to 63 % (Helheim Glacier) of the available satellite

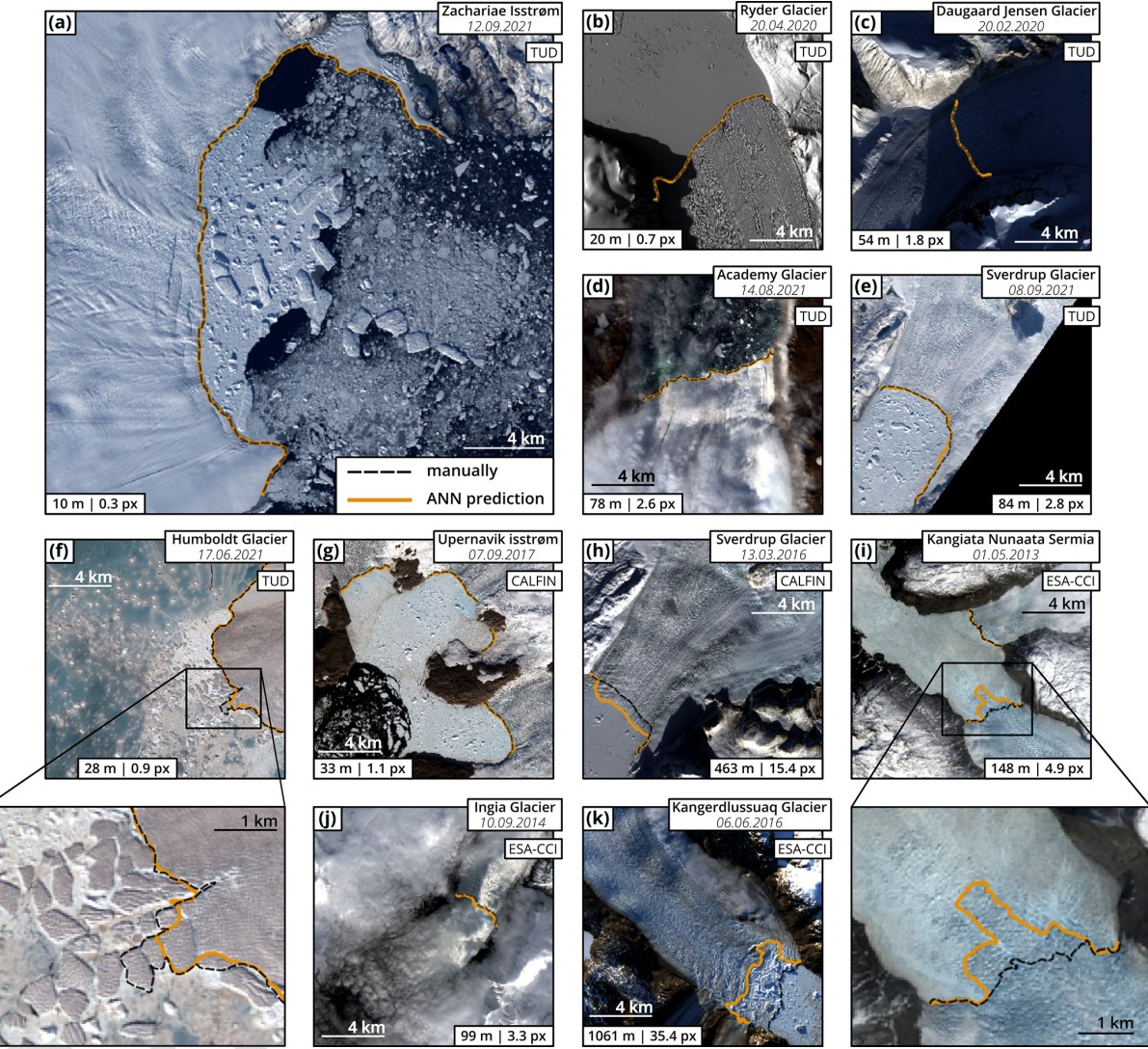

**Figure 3.** Test results for example scenes from the **(a–f)** TUD, **(g,h)** CALFIN, and **(i–k)** ESA-CCI testing sets. Manually delineated calving fronts are depicted as dashed black lines. The ANN prediction is shown in orange. The average minimal-distance error for the respective scene is given both in meters and pixels. All depicted results are from the same fitted ANN model. Landsat-8 imagery is courtesy of the U.S. Geological Survey.

**Table 1.** Results of the accuracy assessment. The average minimal distance and Hausdorff distance are provided for the TUD, ESA-CCI, and CALFIN test sets. For both estimates, we provide mean and median values. The standard deviations are calculated from the 50 different models. TS3

| Test dataset | Average minimal distance | | Hausdorff distance | |
|---|---|---|---|---|
| | Mean (m) | Median (m) | Mean (m) | Median (m) |
| TUD | $61.2 \pm 7.5$ | $28.3 \pm 1.4$ | $283.9 \pm 28.1$ | $156.4 \pm 7.2$ |
| ESA-CCI | $73.7 \pm 2.9$ | $45.9 \pm 1.4$ | $352.4 \pm 14.1$ | $205.4 \pm 10.3$ |
| CALFIN | $73.5 \pm 3.3$ | $43.6 \pm 1.6$ | $233.9 \pm 5.7$ | $162.9 \pm 4.8$ |

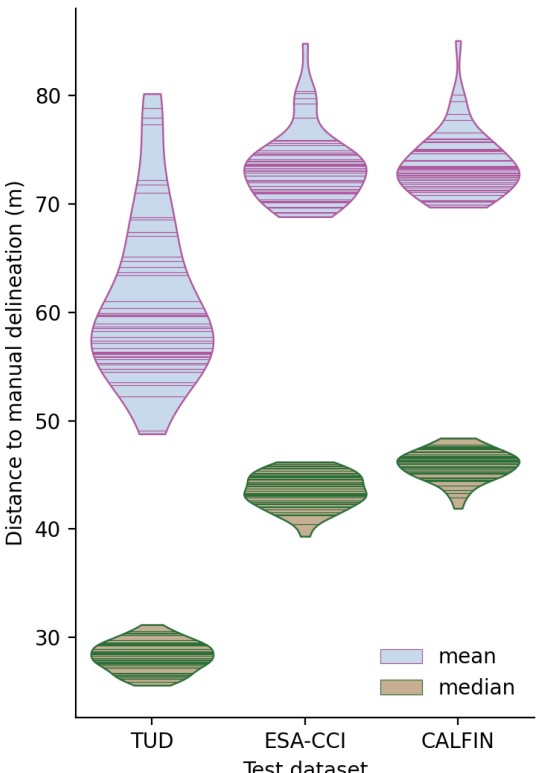

**Figure 4.** Accuracy assessment for the three independent test datasets. Each horizontal line inside the violin plots represents 1 of the 50 trained models applied to the test dataset. The vertical extent of each plot is defined by the corresponding minimum and maximum values.

scenes are discarded before downloading begins. After ANN processing, failed calving front extractions are discarded. Calving front extraction fails when the predicted coastline trajectory does not intersect the static mask. Finally, calving fronts are filtered using the time series. To achieve this, we separate all entries with an area difference larger than 1 km$^2$ from both the previous and the next entry. Separated entries are checked manually. Out of the 10 587 satellite scenes processed by our ANN, 1344 calving front predictions (13 %) were discarded. Figure 6 provides a tabular overview of the final data product (for locations, see Fig. 1). In total, we provide 9243 calving front lines, with most sampling occurring at subweekly intervals outside of polar night. Due to overlapping satellite orbits, glaciers in northern, northeastern, and northwestern Greenland underwent up to six image acquisitions per week depending on the weather and season. Since we use optical data in this study, our time series exhibit observation gaps during polar nights. Depending on the latitude, this gap lasts about 1 month for glaciers in southern Greenland and up to 3 months for glaciers in northern Greenland.

### 4.2 Long-term, seasonal, and subseasonal calving front changes

Marine-terminating glaciers experience calving front variations at different timescales. While long-term changes are easy to resolve using already available data products, our time series offers unique opportunities to analyze seasonal and subseasonal terminus changes. To quantify these calving front changes, we apply the well-established rectilinear-box method (Moon and Joughin, 2008). Rather than using a single profile to measure advance or retreat, this method adopts a rectilinear box, accounting for uneven changes along the calving front. Figure 7 shows the method applied to our calving front time series for the Jakobshavn Isbræ, which is separated into a northern branch and a southern branch. The inferred calving front variation exhibits a pronounced annual pattern combined with smaller subseasonal fluctuations. For comparison, the derived time series of the manually delineated ESA-CCI product is shown. Although both datasets agree very well when it comes to comparing singular epochs, the ESA-CCI time series does not reliably capture the temporal variations. This is particularly evident for the year 2014, when the manually delineated product failed to capture an entire annual cycle.

Figure 8 presents 12 more examples of our ANN-generated time series. Most of these glaciers exhibit pronounced seasonal and subseasonal variations overlaid by a long-term signal. Apart from Kangiata Nunaata Sermia (Fig. 8a), the Ryder Glacier (Fig. 8b), and the Hayes Glacier (Fig. 8h), all example glaciers retreat during the analyzed time period. Notably, the Zachariae Isstrøm and Humboldt Glacier show an area loss of about 120 and 100 km$^2$, respectively. The Ryder Glacier (Fig. 8b) and Nioghalvfjerdsbræ (Fig. 8d) are the only glaciers among the 23 in our study that do not exhibit pronounced seasonality. In these cases, the calving front variation is characterized by a steady advance and the sporadic detachment of large, kilometer-sized icebergs. The date of detachment is precisely pinpointed by the time series. For the Nioghalvfjerdsbræ (Fig. 8d), the time series also resolves two separate break-offs that occurred in close succession. Other glacier time series, such as those for the Hayes Glacier (Fig. 8h), Tracy Glacier (Fig. 8j), Døcker Smith Glacier (Fig. 8k), Harald Moltke Bræ (Fig. 8l), reflect changes in the calving rate during our observation period. For the Harald Moltke Bræ (Fig. 8l), the onset of this calving front retreat, starting in 2019, coincides with the end of its 6-year-long surging phase, as anticipated by Müller et al. (2021).

### 4.3 Comparison to existing data products

In addition to the dataset produced in this study, there are two other automatic-delineation products with circum-Greenland coverage: the CALFIN dataset from Cheng et al. (2021) and the AutoTerm repository from Zhang et al. (2023). Addition-

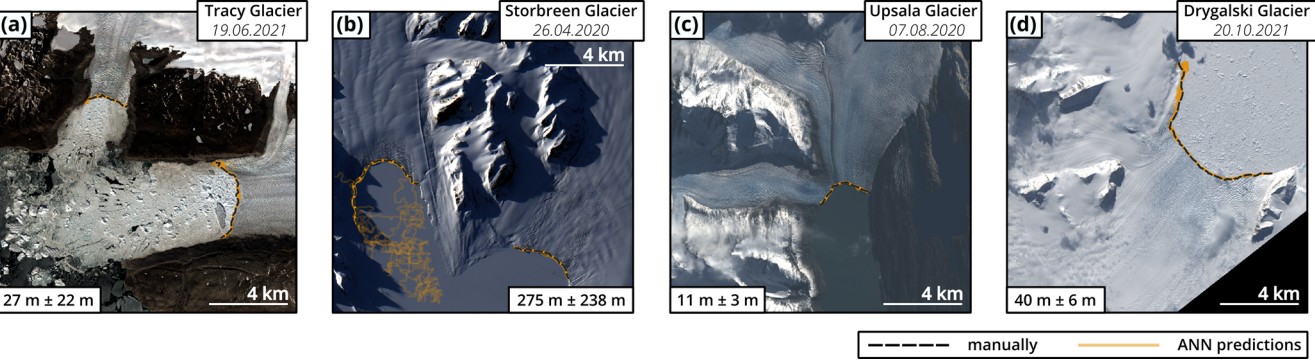

**Figure 5.** Test results for example glaciers which are outside the training dataset – specifically, **(a)** the Tracy Glacier in Greenland, **(b)** the Storbreen glacier in Svalbard, **(c)** the Upsala Glacier in Patagonia, and **(d)** the Drygalski Glacier in Antarctica. Orange lines show the predictions from our 50 models, with overlapping lines indicated by increased color intensity. The average minimal-distance metric for each scene is given in meters. Landsat-8 imagery is courtesy of the U.S. Geological Survey.

| | Kangiata Nunaata Sermia | Helheim Glacier | Kangerdlussuaq Glacier | Jakobshavn Isbræ | Sermeq Avangnardleq | Store Glacier | Rink Isbræ | Daugaard Jensen Glacier | Ingia Isbræ | Upernavik Isstrøm | Waltershausen Glacier | Hayes Glacier | Sverdrup Glacier | Kong Oscar Glacier | Docker Smith Glacier | Harald Moltke Bræ | Tracy Glacier | Humboldt Glacier | Zachariae Isstrøm | Nioghalvfjerdsbræ | Hagen Bræ | Academy Glacier | Ryder Glacier | |
|---|---|---|---|---|---|---|---|---|---|---|---|---|---|---|---|---|---|---|---|---|---|---|---|---|
| 2013 | 5 | 13 | 19 | 15 | 13 | 18 | 20 | 13 | 18 | 6 | 13 | 13 | 14 | 9 | 4 | 16 | 19 | 7 | 15 | 20 | 51 | 54 | 43 | 4 |
| 2014 | 23 | 33 | 42 | 29 | 33 | 40 | 43 | 30 | 30 | 19 | 18 | 33 | 38 | 26 | 25 | 41 | 46 | 32 | 46 | 38 | 66 | 60 | 53 | |
| 2015 | 19 | 24 | 29 | 29 | 32 | 39 | 45 | 26 | 33 | 40 | 25 | 42 | 53 | 44 | 42 | 51 | 58 | 36 | 42 | 40 | 118 | 128 | 100 | |
| 2016 | 17 | 29 | 38 | 26 | 29 | 37 | 44 | 38 | 44 | 33 | 28 | 46 | 48 | 38 | 42 | 51 | 54 | 45 | 60 | 62 | 127 | 127 | 105 | |
| 2017 | 13 | 27 | 39 | 25 | 32 | 36 | 40 | 30 | 41 | 36 | 29 | 40 | 46 | 44 | 38 | 50 | 47 | 38 | 68 | 56 | 126 | 112 | 88 | |
| 2018 | 17 | 17 | 40 | 25 | 35 | 40 | 37 | 29 | 38 | 33 | 39 | 46 | 45 | 34 | 37 | 52 | 54 | 40 | 60 | 50 | 135 | 89 | 114 | |
| 2019 | 15 | 35 | 47 | 34 | 37 | 43 | 46 | 42 | 49 | 46 | 46 | 64 | 60 | 48 | 57 | 54 | 51 | 48 | 73 | 62 | 103 | 99 | 98 | |
| 2020 | 13 | 29 | 40 | 25 | 31 | 35 | 37 | 28 | 47 | 43 | 30 | 40 | 56 | 44 | 53 | 52 | 58 | 47 | 52 | 48 | 93 | 82 | 107 | Entries |
| 2021 | 23 | 31 | 43 | 30 | 29 | 33 | 52 | 56 | 44 | 37 | 41 | 46 | 59 | 48 | 44 | 50 | 49 | 33 | 42 | 64 | 116 | 108 | 86 | 135 |
| Total | 145 | 238 | 337 | 238 | 271 | 321 | 364 | 292 | 344 | 293 | 269 | 370 | 419 | 335 | 342 | 417 | 436 | 326 | 458 | 440 | 935 | 859 | 794 | 9243 |

**Figure 6.** Temporal coverage of our ANN-generated time series. The numbers and color intensity indicate the number of processed calving front positions in each year. Glaciers are sorted by latitude from south (left) to north (right).

ally, there are a number of manually picked data records. Two particularly comprehensive databases are the TermPicks product (Goliber et al., 2022) and the dataset from Black and Joughin (2023). The TermPicks product (Goliber et al., 2022) is a compilation of manually delineated calving front data from 19 different authors across 278 glaciers. The Black and Joughin (2023) dataset was created to study weekly and monthly calving front variability. For this purpose, the authors digitized 199 glaciers with a monthly frequency and 20 glaciers with a 6 d frequency over a 7-year period. In this section, we will compare these four big-data calving front datasets on Greenland's glaciers with the results of this study. This comparison takes place on three levels. Firstly, we compare the general statistics and scope. Secondly, we compare results over a reference period and reference glaciers that are defined according to their temporal and spatial overlap. Thirdly, we examine individual examples.

Table 2 (second through fourth columns) presents the general statistics for the four datasets. Our dataset covers a relatively short time span since we process imagery from the OLI and TIRS Landsat sensors, which have only been active since 2013. With 9243 mapped calving fronts over 23 glaciers, our data product is smaller in terms of both scope and size than the CALFIN, AutoTerm, TermPicks, and Black and Joughin (2023) products. When examining the number of calving front traces, it is important to understand that the definition of what a single calving front contains varies from study to study. For instance, a single data entry in our dataset for the Upernavik Isstrøm includes four calving front features. The CALFIN product lists three separate calving fronts for the same glacier, and the AutoTerm and TermPicks products list two. For the Jakobshavn Isbræ, the CALFIN product considers the northern and southern branches separately, while in our dataset, they are counted as one calving front. In addition, some of our predictions include smaller neighboring glaciers that are located on the same image tile (e.g., the Farquhar Glacier, included with the Tracy Glacier, or Akullersuup Sermia, included with Kangiata Nunaata Sermia). Usually, this applies to glaciers in a single glacier system that were previ-

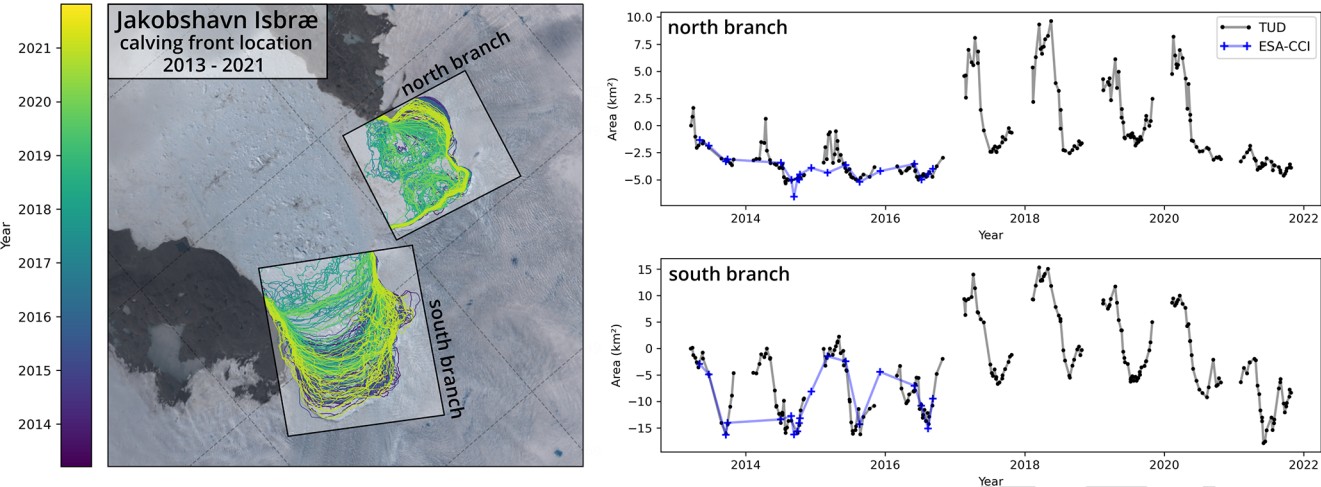

**Figure 7.** Rectilinear-box method applied to the ANN-generated calving front time series for the Jakobshavn Isbræ (western Greenland). The glacier, which is separated into a northern branch and a southern branch, and the calving fronts are shown on the left. The corresponding time series are depicted on the right. Here, calving front positions, expressed as surface areas, are marked with dots. For the TUD product (black), solid lines connect frontal positions for each year. Time series from the ESA-CCI product (blue) are shown for comparison. Landsat-8 image courtesy of the U.S. Geological Survey.

ously connected. When counting shapefile features, the number of entries in our data product is 15 150.

To better compare the data products and differences in processing strategies, we define a reference period and reference glaciers by considering the temporal overlap (2015 to 2019) and spatial overlap (13 glaciers) of the four datasets. These glaciers are Kangiata Nunaata Sermia, the Helheim Glacier, the Kangerdlugssuaq Glacier, the Jakobshavn Isbræ, the Sermeq Avangnardleq glacier, the Store Glacier, the Rink Isbræ, the Ingia Isbræ, the Upernavik Isstrøm, the Hayes Glacier, the Sverdrup Glacier, the Kong Oscar Glacier, and the Døcker Smith Glacier. Within this reference period, we analyzed mapped fronts, sampling rates, and unique entries. Results are provided in Table 2 (fifth to seventh columns). We consider only one calving front entry per day and per glacier. Effectively, this removes (1) duplicate delineations of the same scene (e.g., from multiple authors in the TermPicks database or from the reference data included in the CALFIN dataset) and (2) inconsistencies in what constitutes a single calving front entry. Although based on the same Landsat data, our data product achieves a higher sampling rate and more unique front extractions than the CALFIN product. This is likely due to differences in input feature selection and processing. The AutoTerm product has the most mapped and unique fronts as well as the highest sampling rate. This is mainly due to its ability to process multisensor imagery and its resulting larger database, which includes Landsat, Sentinel-2, and Sentinel-1 data. This provides a clear advantage over our approach, which is limited to the use of multispectral Landsat data. The manually picked TermPicks and Black and Joughin (2023) datasets have a lower (yet still comparable) sampling rate to that of our product. It should

also be noted that the sampling rate of all five calving front products, and thus the number of mapped fronts, is unevenly distributed across the glaciers. This is due to varying satellite image availability and quality, and for manually digitalized products, it is also due to time constraints and prioritization. This is evident not only in the TermPicks database, which has a significantly higher sampling rate in western Greenland than in eastern Greenland (Goliber et al., 2022), but also in the Black and Joughin (2023) dataset, where 8 of the 13 glaciers in our reference period have a 6 d sampling rate, while the remaining glaciers have a monthly sampling rate. Overall, our method achieves the second-highest sampling rate within this reference period, with 281 out of the 2423 extracted calving fronts not extracted by the CALFIN, AutoTerm, Black and Joughin (2023) or TermPicks products.

Figure 9 shows the time series of the CALFIN, AutoTerm, Black and Joughin (2023), and TermPicks products compared to our study for four individual glaciers. To maximize the sampling of the different datasets, we analyze the central line profiles instead of using the box method. The mean distance for same-day calving front acquisitions ($d$) is indicated for each pair of time series. When examining these four examples, we observe generally good agreement between the time series. Significant differences exist only for the Humboldt Glacier (Fig. 9d). Here, the data quality of the AutoTerm product seems to be notably worse than for the other glaciers, with large fluctuations of up to 5 km in distance. This may be attributed to the large size of the glacier's front, which, at least when using our method, required additional processing steps. For Kangiata Nunaata Sermia (Fig. 9b), our data product is the only one which captures the signal from the seasonal ice tongue (Motyka et al., 2017; Moyer

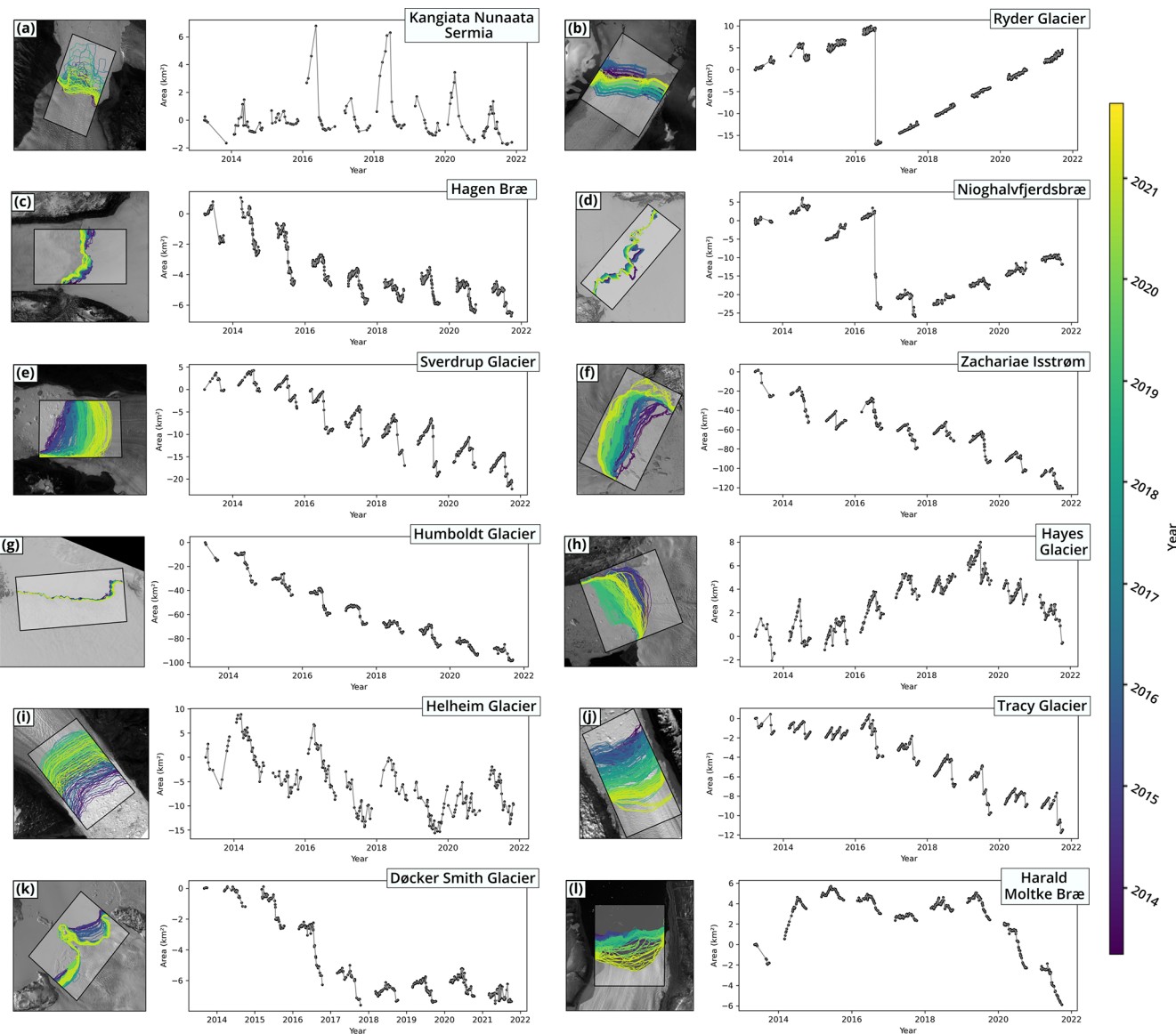

**Figure 8.** Example time series generated by our ANN algorithm for 12 glaciers in Greenland. Each panel illustrates a glacier, with a satellite image showing the color-coded calving front trajectories on the left and the corresponding time series on the right. Here, calving front positions are marked with black dots, and solid lines connect the entries for each year. Note that the $y$ axis is scaled differently in each panel. Landsat-8 imagery is courtesy of the U.S. Geological Survey.

**Table 2.** Comparison of the CALFIN, AutoTerm, Black and Joughin (2023), and TermPicks products with the data product presented in this study. The reference period (2015 to 2019) and the reference glaciers (13 glaciers) are defined by the temporal and spatial overlap of the four data products.

| Dataset | Glaciers | Mapped fronts | Time span | Reference period and glaciers | | |
|---|---|---|---|---|---|---|
| | | | | Mapped fronts | Sampling rate ($yr^{-1}$) | Unique entries |
| This study (Loebel et al., 2023) | 23 | 9243 | 2013–2021 | 2423 | 37.28 | 281 |
| CALFIN (Cheng et al., 2021) | 66 | 22 678 | 1972–2019 | 956 | 14.71 | 3 |
| AutoTerm (Zhang et al., 2023) | 295 | 278 239 | 1984–2021 | 6512 | 100.18 | 2545 |
| Black and Joughin (2023) | 219 | 23 333 | 2015–2021 | 2187 | 33.65 | 676 |
| TermPicks (Goliber et al., 2022) | 278 | 39 060 | 1916–2020 | 1806 | 27.78 | 271 |

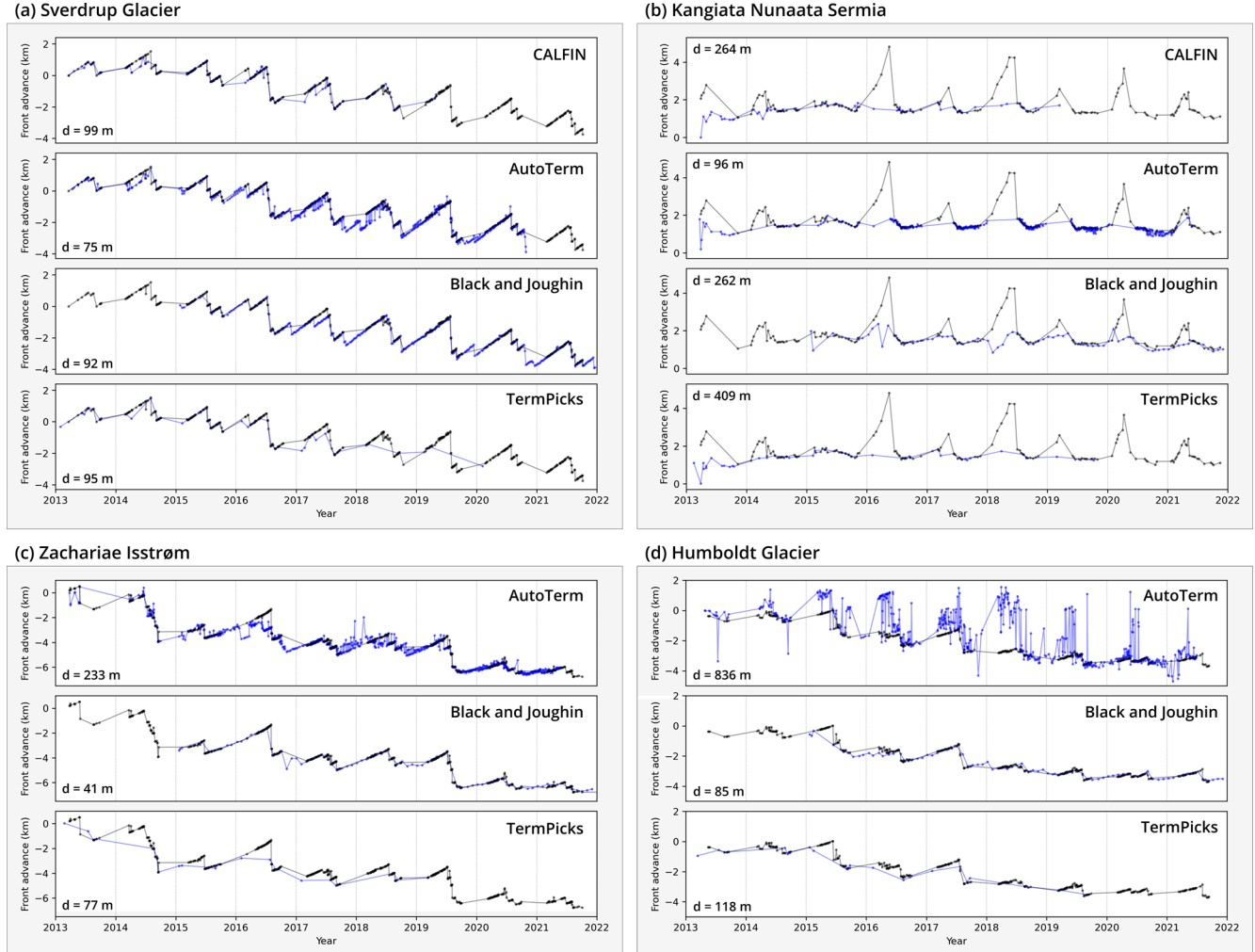

**Figure 9.** Comparison of the CALFIN, AutoTerm, Black and Joughin (2023), and TermPicks products (blue) with the data product presented in this study (black) for four example glaciers. Time series are derived along the central flow line for each glacier. Each comparison specifies the mean distance $d$ between the calving front delineations on the same days.

et al., 2017). This is reflected in the gaps in the other datasets as well as in the greater distances for certain same-day acquisitions. Although Landsat imagery is available, both the CALFIN and AutoTerm products exhibit almost no calving front traces during the emergence, presence, and disintegration of this seasonal ice tongue. We suspect that the multispectral input information in our processing leads to a better extraction rate for scenes under these challenging conditions. All four examples highlight the varying sampling rates of the data products. In particular, the AutoTerm and Black and Joughin (2023) datasets, which use Sentinel-1 SAR imagery, exhibit coverage during polar nights, as observed in late 2017 (Fig. 9a) and late 2016 (Fig. 9c). The sampling rate of the TermPicks repository is lower than that of the automated processing systems shown in these four examples. Compared to the CALFIN, AutoTerm, Black and Joughin (2023), and TermPicks products, our data product has the lowest coverage and smallest number of overall mapped calving front traces. However, due to different processing and the addition of multispectral input information, our method is able to extract a significant number of calving fronts that could not be extracted using the other methods (i.e., 12 % within the reference period). Importantly, these calving fronts (amounting to 12 %) are likely to include extractions under challenging ice mélange and illumination conditions. For the analyzed reference period, our method has a temporal resolution that is second only to that of the AutoTerm product, which benefits from multisensor input imagery. Overall, this comparison also presents a clear argument for the benefits of having multiple data products for monitoring glacier calving fronts. Current data products not only differ in scope but also differ in regard to duplicate extractions for identical glacier front traces, which often exceed estimated delineation uncertainties. A better understanding

of these differences is crucial and requires further investigation. As a final point, we want to emphasize the potential of combining different glacier front products (Goliber et al., 2022). Particularly for datasets based on optical data, this not only increases the overall sampling rate but also allows for data gaps to be filled during the polar winter. Greene et al. (2024) have demonstrated the advantages of such a combination for large-scale glaciological analyses.

## 5  Discussion

Changes in calving front position are, along with other observables (such as ice velocity and elevation change), part of a complex feedback cycle between a glacier and its environment. Long-term calving front trends exhibited by glaciers in Greenland are well characterized (Howat and Eddy, 2011; King et al., 2020; Fahrner et al., 2021; Black and Joughin, 2022; Greene et al., 2024). However, about 80 % of Greenland's glaciers also experience terminus changes on seasonal and subseasonal bases (Black and Joughin, 2023). A visual inspection of the time series shows that 19 of the 23 glaciers analyzed in this study exhibit a seasonal pattern between the years 2013 and 2021. As observed in other studies (Joughin et al., 2008b; Seale et al., 2011; Carr et al., 2013; Schild and Hamilton, 2013; Murray et al., 2015; Moon et al., 2015; Cassotto et al., 2015; Kehrl et al., 2017; Fried et al., 2018; Sakakibara and Sugiyama, 2020; Kneib-Walter et al., 2021; Black and Joughin, 2023), glacier retreat typically starts in late spring, with retreat rates peaking in late summer. A number of mechanisms have been identified as controls for these seasonal terminus changes. These include the duration and timing of meltwater runoff (Sohn et al., 1998; Nick et al., 2010; Chauche et al., 2014; Carroll et al., 2016; Fried et al., 2018; Wood et al., 2021), changes in buttressing force due to sea ice and ice mélange (Howat et al., 2010; Carr et al., 2013; Todd and Christoffersen, 2014; Cassotto et al., 2015; Moon et al., 2015; Kehrl et al., 2017; Robel, 2017; Kneib-Walter et al., 2021), basal sliding (De Juan et al., 2010; Moon et al., 2015), and ocean-driven melt (Motyka et al., 2003; Bevan et al., 2012a; Chauche et al., 2014; Carroll et al., 2016). When a glacier is forced into a state of retreat, both the rate and pattern of retreat are modulated by the subglacial topography. For marine-terminating glaciers in Greenland, this effect has been studied (Warren, 1991; Warren and Glasser, 1992; Joughin et al., 2008b; Carr et al., 2015; Lüthi et al., 2016; Kehrl et al., 2017; Bunce et al., 2018; Catania et al., 2018; Felikson et al., 2021) and modeled intensively (Enderlin et al., 2013; Morlighem et al., 2016; Choi et al., 2017). In particular, faster retreat rates have been found to be associated with overdeepening and retrograde topography.

The new generation of automatically delineated calving front data products not only facilitates glaciological analysis through significant time savings but also potentially provides new insights due to their high temporal resolution and spatial coverage.

Figure 10 shows our calving front time series in relation to bedrock elevation data, taken from the fifth version of the BedMachine Greenland model (Morlighem, 2022), for three example glaciers. Profiles extend from point A to point B along a central flow line. The calving front of the Ingia Isbræ (Fig. 10a) retreated by 3.2 km (8.6 km$^2$ in area) from 2013 to the end of 2017, revealing a pronounced seasonal pattern. Due to retrograde topography (at ∼ 4 km in Fig. 10a), this retreat was particularly rapid in 2016 and 2017. Since 2018, the calving front has been at a topographic minimum, preventing further retreat and reducing the seasonal amplitude. These observations confirm the analysis by Catania et al. (2018), which describes the continuous retreat of the Ingia Isbræ from 2002 to 2016 and suggests further retreat by more than 1 km until the calving front stabilizes on the prograde bed topography. The calving front changes exhibited by the Kangerdlugssuaq Glacier (Fig. 10b) show high seasonal amplitudes as well as a significant retreat from 2016 to early 2018. With the exception of 2017 and 2018, where we observe a sustained retreat, the seasonal amplitude remains almost constant at around 4 km (21 km$^2$ in area). This calving front pattern is also described by Kehrl et al. (2017). Furthermore, the authors show that the Kangerdlugssuaq Glacier's grounding line retreated in 2010 and 2011 to a stable bedrock position (bedrock bump at ∼ 10 km in Fig. 10b), resulting in a floating ice tongue of ∼ 5 km in length. Due to the retrograde bedrock topography located after the bedrock bump (from 10 to 15 km in Fig. 10b) and the reinitialization of the seasonal terminus pattern from 2019 to 2021, we suspect that the calving front retreat from 2016 to 2018 was coupled with a grounding-line retreat on retrograde bed topography, followed by restabilization further inland (likely at 15.5 km, as shown in Fig. 10b). This would confirm the second scenario suggested by Brough et al. (2019). The calving behavior of the Daugaard-Jensen Glacier (Fig. 10c) is influenced by the abrupt change in the bedrock slope close to the frontal position. An advance beyond this point, from a slightly retrograde to a steeply prograde topography, leads to a loss of basal drag and longitudinal stresses. This influences calving behavior and particularly favors the calving of tabular icebergs, as seen in 2013 (8.2 km$^2$ in size) and 2020 (4.2 km$^2$ in size). Although the front of the Daugaard-Jensen Glacier has remained roughly in this location for over 70 years (Stearns et al., 2005) and is considered to be stable (Bevan et al., 2012b), high-temporal-resolution calving front information is still necessary to resolve and understand these stable glacier dynamics. More generally, high-temporal-resolution calving front information not only allows us to analyze glacier retreat and advance but also helps us to better differentiate between different calving styles and patterns.

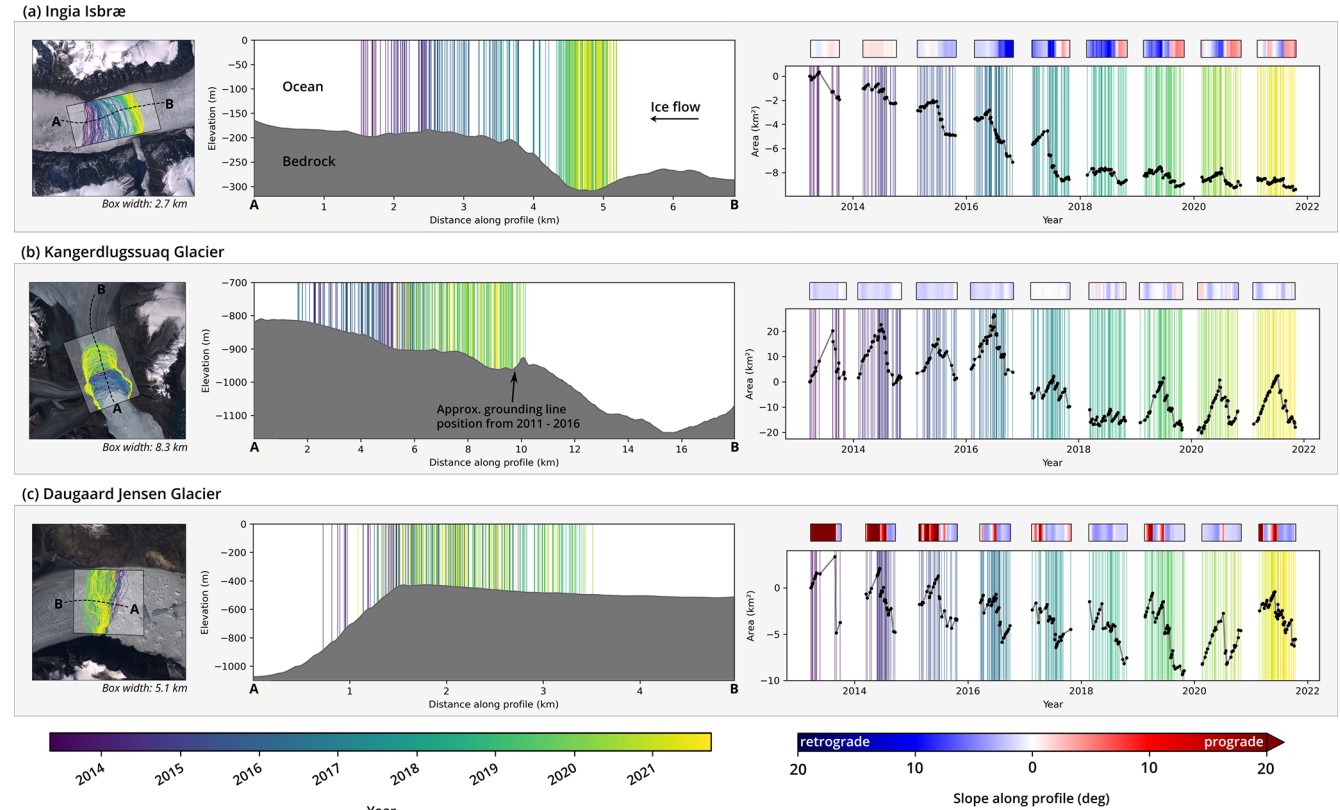

**Figure 10.** Effect of bedrock topography on calving front variations for **(a)** the Ingia Isbræ, **(b)** the Kangerdlugssuaq Glacier, and **(c)** the Daugaard-Jensen Glacier. In each panel (from left to right), there is a satellite image showing calving front trajectories, a marked profile indicating bedrock topography, color-coded calving front positions along the profile, and the corresponding time series of calving front variations. Note that the axes are scaled differently in each panel. Landsat-8 imagery is courtesy of the U.S. Geological Survey.

## 6 Conclusions

This study presents a deep-learning-based processing system for the automatic delineation of calving front locations from multispectral Landsat-8 imagery. Using three independent test datasets, we validate the performance and spatial transferability of our processing system. The quality of the automatically extracted calving fronts is comparable to that of manually delineated calving fronts. Our method achieves a considerably higher extraction rate compared to that of other automation methods that are based on the same Landsat data. The resulting data product, which includes 9242 calving fronts over 23 glaciers within Greenland, is therefore a valuable contribution to the existing data repositories. The presented method and the resulting data product address the needs of the glaciology community for a comprehensive parameterization of glacier calving in Greenland. The time series derived from this processing system resolve long-term, seasonal, and subseasonal calving front variations. This benefit is particularly significant with regard to large glaciers for which manually delineated data are lacking, such as the Humboldt Glacier, Zachariae Isstrøm, and Nioghalvfjerdsbrae. Due to the spatial transferability of this

method, our processing system has the potential to be applied to other marine-terminating glaciers around the world. By presenting the time series in this paper, we offer only a selected glimpse into the dynamics of these glaciers. However, the demonstrated capability of automatically resolving subseasonal calving front variations is an important step towards having a spatially comprehensive Greenland-wide monitoring system. In conjunction with other components concerning ice flow, elevation change, solid-Earth response, and hydrological processes, this will open up new opportunities to holistically assess, model, and simulate dynamic ice sheet changes. Advancing towards this digital twin of the Greenland Ice Sheet will improve our understanding of its evolution and its role within the broader Earth climate system. Intelligent processing strategies, such as deep ANNs, will play a major role in shaping the future of glacier monitoring and associated modeling tasks. This is especially true for analyzing the increasing amount of remote sensing imagery. Well-trained and thoroughly validated ANNs will become the state-of-the-art method for automated calving front delineation. The results presented in this paper will contribute to future advancements in this field.

*Code and data availability.* The following assets are published along with this article:

- The data product featuring automatically delineated calving front positions (Environmental Systems Research Institute (ESRI) shapefile format), which contains 9243 calving front positions across 23 outlet glaciers within Greenland, is available at https://dx.doi.org/10.25532/OPARA-208 (Loebel et al., 2023).

- All reference data applied in this study are available at https://dx.doi.org/10.25532/OPARA-282 (Loebel et al., 2024). This includes 898 manually delineated calving front positions provided in a georeferenced shapefile format as well as 1220 machine-learning-ready preprocessed raster subsets (nine channels) along with their corresponding manually delineated segmentation masks.

- We provide a containerized implementation of the presented processing system using the platform Docker. The software automatically extracts calving front positions from Landsat-8 or Landsat-9 Level-1 data archives for glaciers used in this study or at user-defined coordinates. This enables the analysis of glaciers that are outside our reference dataset or beyond the temporal frame of our study. The software is available at https://github.com/eloebel/glacier-front-extraction (last access 24 March 2023) and https://doi.org/10.5281/zenodo.7755774 (Loebel, 2023a).

- Our implementation of the rectilinear-box method, developed using Python 3, is available at https://github.com/eloebel/rectilinear-box-method (last access 24 March 2023) and https://doi.org/10.5281/zenodo.7738605 (Loebel, 2023b).

*Supplement.* The supplement related to this article is available online at: https://doi.org/10.5194/tc-18-1-2024-supplement.

*Author contributions.* Contributions are listed in accordance with CRediT (Contributor Roles Taxonomy). Conceptualization: EL, MS, and AH. Data curation: EL and CL. Formal analysis: EL. Funding acquisition: MS, MH, AH, and XXZ. Investigation: EL. Methodology: EL and KH. Software: EL. Supervision: MH and MS. Validation: EL. Visualization: EL. Writing (original draft): EL. Writing (review and editing): EL, MH, MS, KH, AH, CL, XXZ, and JS.

*Competing interests.* The contact author has declared that none of the authors has any competing interests.

ther geographical representation in this paper. While Copernicus Publications makes every effort to include appropriate place names, the final responsibility lies with the authors.

*Acknowledgements.* We would like to thank the USGS for providing the Landsat imagery. Additionally, we are grateful to the Center for Information Services and High Performance Computing (ZIH) at TU Dresden for providing their high-performance computing and storage infrastructure. We acknowledge the QGreenland package from the National Snow and Ice Data Center. Finally, we thank the three anonymous reviewers and the editor, Bert Wouters, for their constructive comments, which helped to improve the paper.

*Financial support.* This work was supported by the Helmholtz Association of German Research Centers as part of the Helmholtz Information & Data Science pilot project "Artificial Intelligence for Cold Regions" (AI-CORE; grant no. ZT-I-0016), by the German Federal Ministry of Education and Research (BMBF) project "Greenland Ice Sheet/Ocean Interaction" (GROCE2; grant no. 03F0778G), and by the German Federal Ministry for Economic Affairs and Climate Action TS4 in conjunction with the "national center of excellence ML4Earth" (grant no. 50EE2201C). TS5

*Review statement.* This paper was edited by Bert Wouters and reviewed by three anonymous referees.

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

**Remarks from the language copy-editor**

CE1    Please check that the meaning of your sentence is intact.

CE2    It has been noted that both "Kangerdlugssuaq" and "Kangerdlussuaq", i.e., two different (yet valid) spellings, have been used to refer to this glacier. Please clarify whether or not this inconsistency is okay, and if it is not, please confirm which spelling should be used throughout the paper (both in the figures and text). Thank you.

**Remarks from the typesetter**

TS1    Please send the corrected Supplement and figures along with your next proofreading file. We will exchange them.

TS2    Please give an explanation of why this needs to be changed. We have to ask the handling editor for approval. Thanks.

TS3    Please note that it is not possible to adjust $\pm$ to align further with the vertical line beyond its current position.

TS4    Please confirm this is "Bundesministerium für Wirtschaft und Klimaschutz".

TS5    Please confirm both the Acknowledgements and Financial support sections.

TS6    Please confirm the reference list entry.