# Peer review of "Calving front monitoring at sub-seasonal resolution: a deep learning application to Greenland glaciers"

_The Cryosphere, 2023_

## Author Comment (AC1)

Dear Referee #1,

We thank you for your very useful and constructive review. We will follow your recommendations in the revised version and give detailed answers to your comments below. Some of your points were also raised by the other Referees and we thus refer occasionally also to those answers.

Again, many thanks for helping to improve our manuscript!

Best wishes,

Erik Loebel and all co-authors

**General comments**

The authors present an exciting deep learning method for tracing glacier calving fronts in Landsat 8 and 9 images. The manuscript presents the method based on a specialized Artificial Neural Network, the resulting dataset of 9243 calving front traces for 23 of Greenland's outlet glaciers, and an example of how the data may be useful for examining glacier dynamics. The method produces calving fronts that are on average within 80 meters of manually traced calving fronts, which is less than the uncertainty of manual calving front delineations according to a study by Goliber et al. (2022).

The authors thoughtfully developed the deep learning method. They considered different illumination conditions and terminus morphologies when training the model. There is good documentation of the time and storage requirements for training the model. I applaud the contribution to open-source code and datasets, which are valuable to the glaciological community. The 698 manual delineations used for training the model would also be valuable to the community and I recommend submitting them to a new or existing data repository.

Thank you for your positive and constructive feedback. We fully agree with the recommendation to publish our reference data set. We assume that this data set has considerable value for glaciological analyses and will also prove highly beneficial in subsequent machine learning applications.

In particular this reference data set includes 898 (we used 698 for model training and 200 for model testing) manually delineated calving front positions, which we will provide in a georeferenced shapefile format, as well as 1026 machine learning ready input raster subsets (pre-processed, 9 channels) with their corresponding, manual delineated, segmentation mask. Raster subsets are available in both png and georeferenced tif format. The data is already submitted to the TU Dresden *Open Access Repository and Archive* (OpARA). Reference (with doi) will be included in the revised version.

Overall, the manuscript is well-presented and concisely written. The figures are particularly well-constructed and compelling. However, I think the main text currently lacks detail on the deep learning method. I think the information included in Appendix A and B should be included in the main manuscript since it is relevant to understanding how the ANN algorithm was developed.

This is in line with a comment from Referee #3. We welcome this recommendation and will incorporate appendix A and B into the main body of the manuscript, specifically in Section 2.

Spatial transferability of the method is mentioned throughout the manuscript and described as an advantage to using this method compared to other existing automated calving front tracing methods. The deep learning

model is tested on glaciers from regions outside of Greenland (e.g., Antarctica, Svalbard, and Patagonia). I would be really interested in formal discussion of how the method, trained on Greenland's outlet glaciers, performed with the glaciers in other regions specifically. How does the accuracy calculated for those test glaciers compare to the accuracy of the Greenland test glaciers? Discussing this would provide appropriate support for the spatial transferability of the method.

This comment is much appreciated. When developing our method, we have put a lot of emphasis on model generalization and spatial transferability. In fact, we also process and publish time series of glaciers outside the training dataset.

Although some results are already presented in the figures (e.g. Figure 3 (b) and Figure 7 (j)), we fully agree that a proper discussion of this topic would strengthen the manuscript.

Results for this discussion are already at hand. We calculated the model accuracy separately for (1) glaciers outside the training data set, (2) glaciers outside Greenland and (3) glaciers within the training data set. The results will be presented and comparatively discussed in a new section. In addition, we have created a figure showing example validation results specifically for glaciers from Antarctica, Svalbard and Patagonia (similar in style to Figure 3).

In general, this manuscript presents a valuable contribution to the field and I would like to see this work published after these more major comments and the minor comments listed below are addressed.

**Specific Comments**

In general, proof read for compound adjectives that need to be hyphenated, e.g., Greenland-wide (L176).

Many thanks for the comment. We will double check the text and the compound adjectives it contains.

**L10:** You should include a statement about the accuracy of your method that you calculated here.

The calculated accuracy estimates will be included in the abstract.

**L14-15:** The phrase "digital twin" of Greenland ice sheet is not clearly defined. Unnecessary in abstract unless explained in more detail. It's not discussed throughout the paper so I don't think it's appropriate to include here or in the conclusion without further elaboration.

We agree, the phrase "digital twin" will be removed from the abstract.

**L52-55:** This is not the first automated method that captured sub-seasonal resolution time series of calving front change (see Liu et al., 2021). Reframe the language here.

This point has also been raised by the other referees and we agree. The wording (as well as at L166-168 and L173) will be revised.

**L69:** What is the fixed window size and how was it chosen?

The window size is 512 px by 512 px. Together with the 30 m resolution of Landsat-8, these dimensions were determined to be optimal for capturing the glaciers in Greenland without resampling the imagery. Only Humboldt Glacier, Nioghalvfjerdsbrae and Zachariae Isstrøm do not fit into this window size. This information (some is already in Appendix B) will be included in the main manuscript.

**L72:** "Built" instead of "build"

Will be fixed, thank you.

**L86-100:** This section discussing the method performance should be moved to the Results or Discussion section.

We thank you for this advice. However, this section does not concern the validation of our actual results, but rather the assessment of our machine learning method. The validation of machine learning methods is so closely intertwined with the method itself that we believe it is better to describe these two parts together. Validating our model on independent test data is not a result of our study, but a prerequisite to continuing with our processing and starting with the inference. To make this clear, we propose to rename Section 2.2 from "Validation" into "Accuracy Assessment". For the results and discussion sections, we want to focus on the application to Greenland, the resulting data product, and the glaciological implications. We note that the reviewers had no criticism about this structure. If you strongly prefer a reorganization, please let us know.

**L106:** Elaborate on how the completely clouded Landsat scenes are filtered.

We recognize that the current manuscript lacks detailed information on the filtering of clouded Satellite scenes and failed extractions in our processing workflow. Your comment is therefore very appreciated and also in line with Referee #3. We will rework and expand Section 3 to include information of the following processing steps:

1. Landsat scenes are downloaded using the USGS EarthExplorer (earthexplorer.usgs.gov). Scenes with cloud cover larger than 20% and all *Systematic Terrain Correction* (L1GT) scenes are checked manually. If the glacier front is not visible, the satellite scene is not downloaded for further processing.
2. After the ANN processing, failed calving front extractions are discarded. Calving front extraction fails when the longest feature (which is derived by applying the GDAL contour algorithm with threshold 0.5 on the segmented image) does not intersect the static mask which results in no shapefile being produced.
3. Finally calving fronts are filtered after time series generation using the rectilinear box method. Here we separate all entries with an area difference of larger than 1 km² to both the previous and the next entry. Separated entries are checked manually.

The manual cloud cover check in step 1 was done to reduce download traffic and time. Depending on the glacier 51% (for Ingia Isbræ) to 63% (for Helheim Glacier) of the available satellite scenes are discarded before download. Step 2 and 3 reduced the data product from 10587 to 9243 entries, i.e. discarded about 13% of data.

If data download is no issue (for example when having a local archive), step 1 could be skipped. This will result in significantly more discarded glacier fronts in step 2 and 3.

**L136:** Include citations for how glacier geometry impacts terminus retreat. At the very least, Felikson et al., 2020 (https://doi.org/10.1029/2020GL090112) should be cited here since it directly discusses the impact of bed topography on glacier retreat.

This is a major shortcoming of the current version and the issue has also been raised by the other referees. The discussion will be revised significantly. We will include a literature review on geometric controls on

calving front change and link our results to those of existing studies. Please have a look at our *Reply of RC2* (page 4).

**L140:** Looks more like 2016 and 2017, not 2018 showed the rapid retreat for Ingia Isbræ.

Thank you very much for pointing this out, this will be corrected.

**L164-166:** Is it that the algorithm performs better at overcoming challenging cloud, illumination, and mélange issues than manual delineations? The way this sentence is currently structured implies that. I think this sentence could be removed altogether since the sentence that follows already emphasizes the high temporal resolution of the time series.

We see the problem and appreciate the comment. The sentence will be reworked entirely so it does not imply that the method outperforms manual delineation.

**Figures and Tables**

**Fig. 2.** In the caption, write out TU Dresden or just refer to it as the testing dataset for this study. I think it's fine to exclude the testing glaciers from other regions. Adding a location in parentheses after each of the excluded glaciers would make it more clear why they aren't included in this map. E.g., Drygalski Glacier (Antarctica), Storbreen Glacier (Svalbard), etc.

Many thanks, this is indeed a very good suggestion which we will implement.

**Fig. 5.** I recommend adding a colorbar for the green shading.

We will add a colorbar for Figure 5.

**Fig. B1.** This figure could remain in the Appendix or Supplementary Material even if the description of methodology in Appendix B is moved to main text.

We will move this Figure B1 together with the reworked Table C1 (see comment below) and time series of all other glaciers (not shown in Figure 7) into the supplementary material. Thank you for this suggestion.

**Table C1.** Is the right side really a confusion matrix if only done for TUD? Listing the fraction/percentage of total pixels would be more meaningful here than the raw pixel numbers. As of now, I draw much more from the mean and median errors listed on the left side than the Confusion Matrix. Consider separating the Confusion Matrix portion of this table into its own table. Clearly define TP, TN, FP, FN in the caption.

The confusion matrix is only calculated for the TUD validation set. We have presented it to enable the calculation of commonly used binary classification metrics (like accuracy, F1-Score, recall, precision). Giving the values as percentage is a very good suggestion which we will implement.

We also agree that the distance errors are more meaningful. We will move the table with the distances (which will be expanded to include the mean and median Hausdorf distance, see comment from referee #3) to the main manuscript. The table with the binary classification matrix and the reworked confusion matrix will be moved to the supplement.

---

## Author Comment (AC2)

Dear Referee #2,

First of all we want to thank you for your review and for the constructive feedback on our manuscript. Your general comments in particular are very important points that will lead us to a significantly improved manuscript. Our responses for each of the comments raised and how we addressed them are given below. We would like to apologize in advance for the sometimes lengthy responses. But we believe that especially the results from the comparison with other automatic delineation products make a clear argument on the impact of our study. Some of your points were also raised by the other Referees and we thus refer occasionally to those answers.

Again, many thanks for helping to improve our manuscript!

Best wishes,

**Erik Loebel and all co-authors**

**General Comments**

This paper uses a deep-learning-based method to produce 9243 calving front positions across 23 Greenland outlet glaciers from 2013 to 2021 and discusses the relationship between terminus variation and basal topography. Overall, I think this paper is well-written and the figures are well-presented. With that being said, I have reservations about its originality, impact, and the extent of its literature review. Based on these concerns, I would recommend rejecting the paper in its current form. Below, I provide a detailed evaluation and rationale for my recommendation.

We thank you for your comments and your suggestions for overcoming the deficiencies of the manuscript.

1. Originality:

In recent years, there increasing number of deep learning-based studies to automate terminus extraction. Compared with the previous study, especially with the author's previous paper Loebel et al. (2022), what is the improvement of this study regarding the methodology?

The study *Loebel et al., 2022* focussed on methodology, namely on the selection of input features and their specific contribution to ANN prediction performance for calving front extraction. The results highlight the benefit of multispectral input features as their integration leads to more accurate predictions compared with conventional single band inputs, especially for challenging ice melange and illumination conditions.

This present study builds on the methodological insights from *Loebel et al. 2022*. We apply multispectral Landsat data to process dense calving front time series for Greenland outlet glaciers. Hence, the ANN architecture is the same as in *Loebel et al., 2022* and is not the focus of this manuscript. Apart from the ANN architecture, the present study uses a different set of methods specifically for processing and analyzing time series.

Overall, four key developments extend upon the previous analysis by *Loebel et al., 2022*. First, we improved our processing by limiting input information to multispectral data and by enlarging our training set. Second, we advanced the accuracy assessment by including the ESA-CCI and CALFIN data additionally to our test set (which now also included images from Svalbard and Patagonia). Thirdly, we apply our processing system to automatically delineate 9243 calving front positions and provide these time series for the cryosphere

community. Fourthly, we apply our time series analyzing the interaction of seasonal calving front variation and bedrock topography. Additionally, the revised version will also include a comparison with the other three "big data" repositories (CALFIN, TermPicks and AutoTerm).

We agree that there have been an increasing number of studies advancing deep learning methodology. However, only a limited number of studies (*Cheng et al. 2021, Baumhoer et al. 2023 and Zhang et al. 2023*) exist that apply these methods for time series generation (i.e. applying the method beyond the test dataset), which is ultimately what the cryosphere community requires.

1. Impact:

The main objective of automating the terminus extraction is to produce as many termini as possible. However, compared with the CALFIN (Cheng et al, 2021), which produces 22 678 calving front lines across 66 Greenlandic glaciers from 1972 to 2019, this study seems not improve the temporal resolution, temporal coverage, and spatial coverage. A comparison between the product from this study and CALFIN would be helpful. For instance, which glaciers CALFIN did not cover but this study covers.

We understand this concern. A comparison to other automatic delineation products will not only be highly beneficial for future data users but will also help to emphasize the strengths of our data set and the impact of this study. This is a very good suggestion which has also been raised by the editor and referee #3. The revised version of the manuscript will include a new section presenting and discussing the results of this comparison. In addition to the CALFIN dataset (*Cheng et al. 2021*), we will also analyze the recently published AutoTerm dataset (*Zhang et al. 2023*) as well as the manually delineated TermPicks repository (*Goliber et al. 2022*). The results of this comparison are already at hand and will be briefly introduced below.

Firstly, we want to emphasize that a direct comparison of calving front traces is often problematic since the definition of what a single calving front contains varies from study to study. For example, an entry in our data set for the Upernavik Isstrøm consists of four calving front features whereas in CALFIN these calving fronts are listed separately. For Jakobshavn Isbræ, CALFIN considers the north and south branch separately and in our data set it is one calving front. When adding up shape file features, the number of entries in our data product contains 15150 entries – considerably more than the number of 9243 calving fronts quoted in our manuscript. We will raise and discuss this point in the revised version..

Secondly, we want to emphasize that it is beneficial to have multiple data products on glacier front lines. 10 of our 23 glaciers are not included in the CALFIN product. They include the three large glaciers Humboldt Glacier, Nioghalvfjerdsbrae and Zachariæ Isstøm. This adds up to 8092 (from 9243) traces which are not included in CALFIN. Compared to AutoTerm our data product has 3261 unique traces. Compared to TermPicks, our data product has 8217 unique traces. Compared to the combination of CALFIN, AutoTerm and TermPicks, our data product has 2963 unique traces.

To enable a comparison that also considers the different processing methods, we have set a reference period and reference glaciers by considering their temporal (2013 to 2019) and spatial overlap (13 glaciers). Within this reference we looked at mapped fronts, sampling rate and unique entries. Results are given in Table R1. Although having the same Landsat data basis our data product achieves a higher sampling rate and more unique front extractions than CALFIN. This is due to differences in input feature selection and processing. AutoTerm has the most mapped and unique fronts as well as the highest sampling rate. This is due to its data basis which included Landsat, Sentinel-2 and Sentinel-1. When comparing the results for the TermPicks repository it is important to consider that its sampling rate varies significantly across glaciers. As our reference includes numerous glaciers, which have a relatively large number of entries in the TermPicks

product, the sampling rate here is likely overestimated. Overall, 372 from 3005 of calving fronts extracted by our method within the reference were not extracted by CALFIN, AutoTerm or TermPicks although all use Landsat-8 imagery.

**Table R1:** Tabular comparison of the CALFIN, AutoTerm, TermPicks product as well as the data product presented in this study. The reference period (2013 to 2019) and the reference glaciers (13 glaciers) are defined by the temporal and spatial overlap of the four data products.

| Dataset                          | Glaciers | Mapped fronts | Time span | Reference period and glaciers |                           |                |
|----------------------------------|----------|---------------|-----------|-------------------------------|---------------------------|----------------|
|                                  |          |               |           | Mapped fronts                 | Sampling rate $(yr^{-1})$ | Unique entries |
| This study (Loebel et al., 2023) | 23       | 9243          | 2013-2021 | 3005                          | 33.02                     | 372            |
| CALFIN (Cheng et al., 2021)      | 66       | 22678         | 1972-2019 | 1322                          | 14.53                     | 15             |
| AutoTerm (Zhang et al., 2023)    | 295      | 278239        | 1984-2021 | 7220                          | 79.34                     | 3724           |
| TermPicks (Goliber et al., 2022) | 278      | 39060         | 1948-2021 | 2287                          | 25.13                     | 505            |

We also looked at individual glaciers and compared the data. Figure R1 shows the time series of CALFIN, AutoTerm and TermPicks compared to our study for four examples. Results not only emphasize the different sampling rates, but also highlights the differences in quality control and the associated signal-to-noise ratio. For Kangiata Nunaata Sermia (Fig. R1 (b)) our data product is the only one which captures the seasonal ice tongue (which has been described for example in *Motyka et al., 2017* and *Moyer et al., 2017*).

**Figure R1:** Comparison of the CALFIN, AutoTerm, TermPicks product (blue) as well as the data product presented in this study (black) for four example glaciers. Time series are derived along the central flow line of the glacier. Every comparison specifies the mean distance d of between calving front delineations at identical days.

In Figure R1 we also indicate the mean distance, d, for same-day calving front acquisitions for each pair of time series. On the basis of these differences, some of which extend well beyond estimated delineation uncertainties, it is clear that there are significant and more importantly unexplained differences between different delineation methods. This is a clear argument in favor of having multiple glacier front data products.

Overall, we expect a high impact of this study for two reasons. Firstly, our method enables a considerably higher extraction rate compared to other methods that use the same data basis (in particular compared CALFIN). This results in a significant amount of calving front traces (13% within our reference) that could not be extracted by the other methods. Importantly, these 13% include extractions under challenging conditions such as the appearance and disintegration of the seasonal ice tongue at Kangiata Nunaata Sermia. Secondly, we see enormous value in having multiple calving front data products even if there is a significant overlap. Glacier area change is a designated Essential Climate Variable (ECV, see https://public.wmo.int) and constitutes a basis for glaciological studies and a new generation of ice dynamic models. All current data products are different. They differ in coverage but also differ for duplicate extractions for identical glacier front traces. A better understanding of these differences is crucial and requires further investigation.

We now see that a comparison to previous products is essential for supporting our case, and we thank the referees for raising this important point. The results, which were only briefly presented here, will be discussed in detail in the relevant section of the revised manuscript.

**1. Literature Review:**

The discussion about the terminus variation and basal topography is interesting, but studies have been investigating this for many years (Joughin et al., 2008, 2014; Kehrl et al., 2017; Bunce et al., 2018, Catania et al., 2018), suggesting that retrograde bed slopes can cause glacier dynamic instabilities (Meier and Post, 1987) and substantial retreat. However, these papers are missing in the manuscript. Therefore, I suggest the author include a literature review about how the glacier geometry influences the terminus variation, also remove the expression throughout this manuscript that this study is the first application analyzing the interaction between calving front variation and bedrock topography.

We agree with this comment and similar comments of the other referees. With our discussion we wanted to showcase the potential of our data product for future glaciological analyzes. We agree that such a discussion cannot take place without considering previous work.

The discussion (Section 4) will be substantially revised.

- 1. At the start of the discussion, we will introduce seasonal terminus variability and how it is driven.
- 2. After that, we will discuss geometric controls of terminus change and how different bedrock topography modulates glacier retreat and advance.
- 3. Then, our results will be linked to the results of existing studies. For example, *Catania et al. (2018)* show an ongoing retreat of Ingia Isbræ from 2002 until 2016 and suggest further retreat of more than 1 km inland. Our observations confirm this, and extend this analysis by showing that Ingia Isbræ (Figure 8(a)) has stabilized in 2018 due to the prograde slope.
- 4. We will also emphasize that our time series not only allow to analyze glacier retreat and advance, but also to better differentiate between different calving patterns (e.g. tabular, non-tabular and mixed). Our analysis for Daugaard Jensen (Figure 8 (c)) is a perfect example.
- 5. Throughout the discussion will refer to relevant studies, including all studies mentioned by you and the other referees.

With the phrase "first application" we wanted to refer to a first application of our data product rather than to imply that we are the first to carry out such an analysis. We apologize for this misunderstanding and we will clear this up by rephrasing these expressions.

Considering the points mentioned above, I believe that rejecting the paper would be appropriate. However, I recommend that the authors be given an opportunity to revise and resubmit their work, addressing the concerns mentioned above. By providing constructive feedback and clear expectations for revision, the authors might have the chance to strengthen their manuscript and overcome the present limitations.

**Specific Comments:**

**Line 51:** Why the usage of multispectral sensor information can increase the temporal solution, compared with using the single band? I was asking because using multi-band information could not yield more images.

Using multispectral information does not yield more images, but it does increase the information we have in each image. Different surface types have different reflective curves along the electromagnetic spectrum. This facilitates a better separability between surface types (Kääb et al., 2014). This results in a more robust and accurate image segmentation using ANN which leads to a higher success rate of our method and an increased temporal resolution compared to using single band inputs (Loebel et al., 2022).

The use of multispectral bands is one reason that for the above mentioned reference period our method yields a significantly higher sampling rate compared to CALFIN.

Line 53: I believe the CALFIN can also resolve sub-seasonal terminus variations as it uses all the available Landsat images.

With this sentence we wanted to highlight the ten glaciers which were not included in CALFIN. But we acknowledge that these formulations (also at L166-168 and L173) are problematic, the more so after the recent release of the AutoTerm product. This concern has also been raised by the other referees. The formulations will be reframed

**Line 64:** What modification did the author apply here? Does that modification improve the results? It would be better to have a more detailed description of this.

As a main modification, the present study only uses multispectral information and no textural and topographic features. This reduces the input layer from 17 to 9 layers. Furthermore we increased our TUD reference data set from 728 to 898 calving fronts. These new calving front traces focus specifically on cloudy, low illumination and scene border conditions, enhancing the method in this regard.

We appreciate this comment, and we will expand the revised manuscript with a more detailed description.

**Section 2.2:** It would be better if the author can change the name of this section (validation) to avoid confusion. Following the deep learning convention, there are three sets: the training set, the validation set, and the test set. This section is actually about the test set but is called validation.

Also, the descriptions of the validation set should be included in the manuscript.

We will follow this recommendation and rename section 2.2 to *accuracy assessment*. Furthermore we will follow the suggested convention and throughout the manuscript use the naming: training set, validation set and test set.

A short description of the validation set (in the article *internal validation set*) is already in appendix B and will be moved to section 2 in the main manuscript.

**Figure 8:** It would be more helpful to change the second column to the time series of the bed elevation at the terminus. Time series might better reflect the verbal descriptions in the discussion section.

After looking at the time series of bed elevation at the terminus position, we prefer the current display of profile distance and bedrock depth. Mainly because this lateral view of the glacier and its bedrock topography is very easy and fast to com

---

## Author Comment (AC3)

Dear Referee #3,

We thank you for the constructive comments and the careful assessment on our manuscript. All comments have been taken into account and a list of responses and actions is given below. Many of your points were also raised by the other referees and we thus refer also to those answers. Again, many thanks for helping to improve our manuscript!

Best wishes,

Erik Loebel and all co-authors

**General comments**

The authors present a deep learning method for automating the digitization of glacier calving fronts from Landsat 8 and 9 images, applied to 23 glaciers in Greenland from 2013 through 2021. They demonstrate the potential application of this method in studying aspects of outlet glacier behavior such as calving front seasonality and interactions with bed topography.

Overall the manuscript is clearly written, and the figures are high quality and do a good job of supporting the text. However, I am concerned that it is lacking important context from previous work, and is not sufficiently distinguishable from other automation methods:

First, how does this deep learning method compare to previous contributions to automation, both in methodology and in outcomes (accuracy)? In other words, as a data user, why should I choose the output from this particular automated method over any of the others that have been recently published? The authors highlight the temporal resolution of this dataset, but to me it doesn't appear to be substantially unique from other automated or even some manual datasets.

We thank you for raising this important point. In the current version of our manuscript, this aspect is somewhat neglected. We understand that the comparison with the existing data products in particular is very important for precisely the concerns you mentioned. This was also suggested by the editor and referee #2.

The revised version of the manuscript will include a new section presenting and discussing the results of this comparison. In addition to the CALFIN dataset (*Cheng et al. 2021*), we also analyzed the recently published AutoTerm dataset (*Zhang et al. 2023*) as well as the manually delineated TermPicks repository (*Goliber et al. 2022*). The results of this comparison are already at hand and are briefly introduced at our response to referee #2. Please have a look at our *Reply of RC2* (page 2).

Based on the results of the comparison, we will also highlight differences in processing. A comparison of our ANN architecture with that of CALFIN and many others has already been published in *Heidler et al. (2023)*. Our focus will therefore be on the input data, pre-processing, post-processing and quality control.

Next, for the highlighted seasonal and topographic applications, how do the results presented here agree with or differ from other studies? There is a wealth of literature on both terminus position seasonality and terminus-topography relationships, but none of it is referenced in the presentation of these data applications, and the authors suggest that their analysis is unprecedented, which is simply not true. Both section 3.2 and the discussion would benefit from a more thorough literature review and contextualization of the work presented.

*Thank you for this comment. We agree that the discussion (section 4) needs to be improved. Especially in the points you highlighted. The revised version of our manuscript will include a literature review and contextualisation of our results. This was also pointed out by the other referees. Please see our Reply of RC2 (page 4).*

Finally, can the authors demonstrate that this automated method, and its resultant higher temporal density of calving fronts, improves upon prior analyses? It would be helpful to move beyond demonstrating the capabilities to demonstrating how they are better than existing approaches. Figure 6 starts to get at this with the comparison between TUD and ESA-CCI, but I think this could be explored in more detail.

*That's very well put. As with your first point, we agree. For the revised version we demonstrated that our processing system has a higher sampling rate than CALFIN and than the TermPicks repository. Within this reference we also show that 13% of our extracted calving front traces were not extracted by either CALFIN, TermPicks or the AutoTerm Product.*

*For a more direct comparison we also created Figure R1 (see Reply of RC2 (page 3)). We directly compare our results with CALFIN, TermPicks and AutoTerm for four example glaciers. This figure highlights differences in sampling rate, but also the different approaches in input data and filtering.*

**Specific comments**

L31-35: The authors cite several studies that use manually delineated fronts, then state that these products "often lack temporal resolution, making seasonal analysis … difficult." However, several of the cited products do perform robust seasonal analyses. Schild and Hamilton (2013) had near-daily termini for five glaciers. King et al. (2020) had seasonal data, though centerline positions rather than full calving fronts. Goliber et al. (2022) specifically addressed how their combined dataset could be used to better study seasonality than any of the individual component datasets. Black and Joughin (2023) was explicitly about analyzing sub-seasonal (weekly or monthly) front variability throughout Greenland – in fact, for their weekly dataset, they reported an average of 50 fronts per glacier per year (including polar night), versus 45 in this manuscript (9243 fronts / 23 glaciers / 9 years). While it is true that manual digitization is laborious and will struggle to keep up with the volume of new imagery (L33), that cannot be used to imply that those efforts have not produced comparable datasets. The authors go on to use language such as "unprecedented temporal resolution, resolving [the glaciers'] sub-seasonal calving front variability for the first time" (L52-53), "[this time series] clearly surpasses the potential of manually delineated data products" (L166-168), and "unprecedented temporal resolution" (L173). Papers cited within this manuscript, as well as others, have repeatedly demonstrated comparable temporal resolution of calving front data products, including through manual digitization.

*We agree that this sentence has to be rephrased completely. Manual digitization will always be capable of creating data sets with comparable or better temporal resolution compared to automated delineation, at least in theory. Our argument is (and that's why this sentence has to be changed) about the time-consuming process of manual digitization. We apologize for not making this clear.*

*Many manual delineated data sets are side products of glaciological studies where the authors most likely spend a lot of time on manual digitization. For example, to delineate the calving front data used in Catania et al. (2018), it took the authors approximately 48 hours per glacier (with an average of 367 traces per glacier) (Goliber et al., 2022). That is something we want to change. And there are still many large Greenland glaciers where we don't have enough data to perform seasonal analyzes. Manual delineation did not, and in our opinion never will, keep up with the amount of satellite imagery. The spatially uneven sampling of the*

TermPicks repository (*Goliber et al., 2022*) makes this very clear. Although having much higher satellite revisit times it has significantly less calving front data in north and north-east Greenland than in west Greenland. Our statements were intended to refer to these glaciers (and for L173 explicitly naming Humboldt Glacier, Zachariæ Isstøm and Nioghavfjerdsbrae).

That said, we are very thankful for raising and discussing this point. This issue has also been brought up by the other referees. The formulations (L31-35, L52-53, L166-168 and L173) are not accurate in their current form and we will reframe the language. The phrase "*unprecedented temporal resolution*" will be removed from the manuscript entirely.

L73-75: I'm glad these various messy situations were considered for training data. Did the training data also focus on glaciers that are heavily crevassed near the terminus? This can also be a tricky condition – and different from dense mélange – perhaps wrapped up in "morphological features"?

When assembling the reference data we also made sure that glaciers with heavily crevassed terminus are included. Some examples are shown in Figure 3.

Probably one of most crevassed calving fronts we processed is at Zachariæ Isstrøm. Figure R3 (will not be included in the main manuscript) shows such a predicted calving front. Some of these crevasses get so large that the open water or ice mélange between them becomes visible. A reliable and consistent ANN delineation under these conditions is essential for ensuring an accurate calving parameterization and jump-free time series (see Figure 7 (f) and Figure R1 (in *Reply of RC2)*).

[Figure]

*Figure R3:* *Calving front of Zachariæ Isstøm on 29.04.2016. ANN prediction is shown in orange. The image is not included in the training data set and is therefore unseen by the ANN.*

We will incorporate and explicitly mention the crevassed conditions in the revised text. Thank you for this question.

L78-80: Did this system end up being spatially transferable? The results for looking at the non-Greenlandic glaciers are not addressed.

This point has also been raised by referee #1 (see *Reply of RC1* (page 2)). Although some relevant results are already presented in the figures (e.g. Figure 3 (b) and Figure 7 (j)), we agree that model generalization and spatial transferability has not been addressed sufficiently. Especially, since we put a lot of emphasis on this topic when developing our method.

For the revised manuscript we will discuss spatial transferability in a new subsection in section 2. We will present and discuss model accuracy separately for (1) glaciers outside the training data set, (2) glaciers outside Greenland and (3) glaciers within the training data set. Also we have created a figure showing example validation results specifically for glaciers from Antarctica, Svalbard and Patagonia (similar in style to figure 3).

L98-101: I agree with the overall conclusion that the quality of automated fronts from this method is comparable to the quality of manually delineated termini. The comparison with the error estimate from Goliber et al. (2022) is a helpful reference, but a direct comparison is difficult due to differences in the error calculation method. Goliber et al. (2022) use the Hausdorff distance (they say "greatest minimum distance between two lines") rather than averaging the minimal distances along the front as is done here, so I suspect their error estimate would be a bit larger than what is reported here. It would also be helpful to see how error estimates from this method compare to other automated methods (it looks like this is at least done for CALFIN in Table C1, but this should be mentioned in the main body text about error estimation too).

Thank you for raising this very important point. The distance accuracy estimates are different in almost every study which makes it challenging to directly compare the results. The distance estimate that we use (here: *average minimal distance*) is comparable to the estimates used in CALFIN (*Cheng et al, 2021*) and IceLines (*Baumhoer et al, 2023*).

Incorporating the Hausdorff distance (*Huttenlocher et al., 1993*) will increase comparability to the estimates of *Goliber et al. (2022)* and also improve categorizing our results in general. This is a great idea. Table R2 shows the results for the three testing datasets and will be included in section 2 of the revised manuscript.

*Table R2:* *Results of the accuracy assessment for the TUD, ESA-CCI and CALFIN test set. Both mean and median of the average minimal distance and the Hausdorff distance is given as a mean over 50 model runs.*

| Test datast | Average minimal distance | | Hausdorff distance | |
| --- | --- | --- | --- | --- |
| | Mean (m) | Median (m) | Mean (m) | Median (m) |
| TUD | $61.2 \pm 7.5$ | $28.3 \pm 1.4$ | $283.9 \pm 28.1$ | $156.4 \pm 7.2$ |
| ESA-CCI | $73.7 \pm 2.9$ | $45.9 \pm 1.4$ | $352.4 \pm 14.1$ | $205.4 \pm 10.3$ |
| CALFIN | $73.5 \pm 3.3$ | $43.6 \pm 1.6$ | $233.9 \pm 5.7$ | $162.9 \pm 4.8$ |

As you suspected, the Hausdorff distance is significantly larger than the average minimal distance. Since the Hausdorff distance only considers the greatest distance of all minimum distances along the two trajectories, it is very sensitive towards even small misclassified parts along the predicted calving front. Thus, there is a

bias regarding the length of the calving front. Results will be discussed in the revised version. Incorporating this second distance estimate is a very welcome addition.

L105-106: What percentage of imagery was completely clouded (and therefore unused)?

Thank you for bringing this up. Please see the answer below and our *Reply of RC1* (page 3).

L106-107: What qualifies as a "failed calving front extraction"? What are the criteria for failure? Did the authors manually check all results from the automation?

We agree that the current manuscript lacks detailed information on the filtering of clouded Satellite scenes and failed extractions in our processing workflow. Your comment is therefore very appreciated and also in line with Referee #1. We will rework and expand Section 3 to include all the corresponding information. Please have a look at our *Reply of RC1* (page 3).

Figure 8/Discussion: This is a nice presentation of the relationship between topography and calving fronts over seasons and years. The expanded interpretation in the discussion is helpful, especially for highlighting correlations between topography and seasonal amplitude. However, these interpretations should be placed in the context of other topography-terminus studies (how are they similar/different, what is new here, etc.). Catania et al. (https://doi.org/10.1029/2017JF004499 and others) would be a good starting point.

We agree with this assessment. In the revised version we will link our results to the results of existing studies. As this point was also raised by the other referees, please refer to our *Response to RC2* (page 2).

Appendix B: Since the core of this manuscript is the deep learning method, the methodology should be in the main body rather than tucked away in an appendix.

This point has also been raised by Referee #1. We welcome this recommendation and will incorporate appendix A and B into the main body of the manuscript, specifically in Section 2.

**Technical corrections**

L72: change "build" to "built"

Will be fixed, thank you.

L140: add BedMachine version here (I see v5 in the references, but it is helpful to see up front since there have been some big changes between versions).

The version number will be added in the main text.

L172: remove "are" after Nioghalvfjerdsbræ

Will be removed, thanks.

**References**

Cheng, D., Hayes, W., Larour, E., Mohajerani, Y., Wood, M., Velicogna, I., & Rignot, E. (2021). Calving Front Machine (CALFIN): glacial termini dataset and automated deep learning extraction method for Greenland, 1972–2019. The Cryosphere, 15(3), 1663-1675.

Heidler, K., Mou, L., Loebel, E., Scheinert, M., Lefèvre, S., & Zhu, X. X. (2023). A Deep Active Contour Model for Delineating Glacier Calving Fronts. *IEEE Transactions on Geoscience and Remote Sensing*.

Baumhoer, C. A., Dietz, A. J., Heidler, K., & Kuenzer, C. (2023). IceLines–A new data set of Antarctic ice shelf front positions. *Scientific Data*, *10*(1), 138.

Goliber, S., Black, T., Catania, G., Lea, J. M., Olsen, H., Cheng, D., Bevan, S., Bjørk, A., Bunce, C., Brough, S., Carr, J. R., Cowton, T., Gardner, A., Fahrner, D., Hill, E., Joughin, I., Korsgaard, N. J., Luckman, A., Moon, T., Murray, T., Sole, A., Wood, M., and Zhang (2022). TermPicks: a century of Greenland glacier terminus data for use in scientific and machine learning applications. *The Cryosphere*, *16*(8), 3215-3233.

Zhang, E., Catania, G., & Trugman, D. T. (2023). AutoTerm: an automated pipeline for glacier terminus extraction using machine learning and a "big data" repository of Greenland glacier termini. *The Cryosphere*, 17(8), 3485-3503.

Catania, G. A., Stearns, L. A., Sutherland, D. A., Fried, M. J., Bartholomaus, T. C., Morlighem, M., Shroyer, E., and Nash, J.: Geometric Controls on Tidewater Glacier Retreat in Central Western Greenland, J. Geophys. Res.-Earth, 123, 2024–2038, https://doi.org/10.1029/2017JF004499, 2018.

Huttenlocher, D. P., Klanderman, G. A., & Rucklidge, W. J. (1993). Comparing images using the Hausdorff distance. *IEEE Transactions on pattern analysis and machine intelligence*, *15*(9), 850-863.

---

## Author Response (AR1)

Dear Bert,

Thank you for your work on our manuscript and for inviting us to submit a revised version. We have uploaded this revised manuscript together with a track change version. We have carefully addressed the points raised by the referees and made changes in line with our responses. Below, we also provide definitive answers to how and where the referee's comments were addressed. For comprehensive answers to the questions, please see our responses (Reply on RC1/2/3).

Furthermore, I would like to inform you that I will be on a field campaign from 2 February to 19 April. I apologize in advance for not being able to reply to emails during this period. Our second contact (Martin Horwath) is informed about this, but he will not be able to make any major changes to the manuscript.

Best wishes,

Erik and all co-authors

Anonymous Referee #1, 01 Jun 2023

**General comments**

The authors present an exciting deep learning method for tracing glacier calving fronts in Landsat 8 and 9 images. The manuscript presents the method based on a specialized Artificial Neural Network, the resulting dataset of 9243 calving front traces for 23 of Greenland's outlet glaciers, and an example of how the data may be useful for examining glacier dynamics. The method produces calving fronts that are on average within 80 meters of manually traced calving fronts, which is less than the uncertainty of manual calving front delineations according to a study by Goliber et al. (2022).

The authors thoughtfully developed the deep learning method. They considered different illumination conditions and terminus morphologies when training the model. There is good documentation of the time and storage requirements for training the model. I applaud the contribution to open-source code and datasets, which are valuable to the glaciological community. The 698 manual delineations used for training the model would also be valuable to the community and I recommend submitting them to a new or existing data repository.

All reference data applied in this study is available at http://dx.doi.org/10.25532/OPARA-282. In particular, this includes 898 manually delineated calving front positions provided in a georeferenced shapefile format, as well as 1220 machine learning ready raster subsets (pre-processed, 9 channels) with their corresponding manual delineated segmentation mask.

This is now also referenced in the *Code and data availability* section.

Overall, the manuscript is well-presented and concisely written. The figures are particularly well-constructed and compelling. However, I think the main text currently lacks detail on the deep learning method. I think the information included in Appendix A and B should be included in the main manuscript since it is relevant to understanding how the ANN algorithm was developed.

We have incorporated appendix A and B into the main body of the manuscript, specifically in Section 2.

Spatial transferability of the method is mentioned throughout the manuscript and described as an advantage to using this method compared to other existing automated calving front tracing methods. The deep learning model is tested on glaciers from regions outside of Greenland (e.g., Antarctica, Svalbard, and Patagonia). I would be really interested in formal discussion of how the method, trained on Greenland's outlet glaciers, performed with the glaciers in other regions specifically. How does the accuracy calculated for those test glaciers compare to the accuracy of the Greenland test glaciers? Discussing this would provide appropriate support for the spatial transferability of the method.

Our results on spatial transferability are presented in a new section (section 3.2).

In general, this manuscript presents a valuable contribution to the field and I would like to see this work published after these more major comments and the minor comments listed below are addressed.

**Specific Comments**

In general, proof read for compound adjectives that need to be hyphenated, e.g., Greenland-wide (L176).

Done.

**L10:** You should include a statement about the accuracy of your method that you calculated here.

Done.

**L14-15:** The phrase "digital twin" of Greenland ice sheet is not clearly defined. Unnecessary in abstract unless explained in more detail. It's not discussed throughout the paper so I don't think it's appropriate to include here or in the conclusion without further elaboration.

The phrase "digital twin" is removed from the abstract.

**L52-55:** This is not the first automated method that captured sub-seasonal resolution time series of calving front change (see Liu et al., 2021). Reframe the language here.

The phrasing at this and other locations has been revised.

**L69:** What is the fixed window size and how was it chosen?

This information is now included in the main manuscript (Section 2.3.1).

**L72:** "Built" instead of "build"

Done.

**L86-100:** This section discussing the method performance should be moved to the Results or Discussion section.

The accuracy assessment is now its own section (Section 3).

**L106:** Elaborate on how the completely clouded Landsat scenes are filtered.

Section 4.1 now includes this information.

**L136:** Include citations for how glacier geometry impacts terminus retreat. At the very least, Felikson et al., 2020 (https://doi.org/10.1029/2020GL090112) should be cited here since it directly discusses the impact of bed topography on glacier retreat.

The discussion (Section 5) has undergone significant reworking. It now includes a comprehensive literature review and a more thorough contextualisation of our results.

**L140:** Looks more like 2016 and 2017, not 2018 showed the rapid retreat for Ingia Isbræ.

Fixed.

**L164-166:** Is it that the algorithm performs better at overcoming challenging cloud, illumination, and mélange issues than manual delineations? The way this sentence is currently structured implies that. I think this sentence could be removed altogether since the sentence that follows already emphasizes the high temporal resolution of the time series.

The sentence has been reworked so it does not imply that the method outperforms manual delineation.

**Figures and Tables**

**Fig. 2.** In the caption, write out TU Dresden or just refer to it as the testing dataset for this study. I think it's fine to exclude the testing glaciers from other regions. Adding a location in parentheses after each of the excluded glaciers would make it more clear why they aren't included in this map. E.g., Drygalski Glacier (Antarctica), Storbreen Glacier (Svalbard), etc.

Done.

**Fig. 5.** I recommend adding a colorbar for the green shading.

Done.

**Fig. B1.** This figure could remain in the Appendix or Supplementary Material even if the description of methodology in Appendix B is moved to main text.

Done.

**Table C1.** Is the right side really a confusion matrix if only done for TUD? Listing the fraction/percentage of total pixels would be more meaningful here than the raw pixel numbers. As of now, I draw much more from the mean and median errors listed on the left side than the Confusion Matrix. Consider separating the Confusion Matrix portion of this table into its own table. Clearly define TP, TN, FP, FN in the caption.

The table with the binary classification metrics and the reworked confusion matrix has been moved to the supplement.

**General Comments**

This paper uses a deep-learning-based method to produce 9243 calving front positions across 23 Greenland outlet glaciers from 2013 to 2021 and discusses the relationship between terminus variation and basal topography. Overall, I think this paper is well-written and the figures are well-presented. With that being said, I have reservations about its originality, impact, and the extent of its literature review. Based on these concerns, I would recommend rejecting the paper in its current form. Below, I provide a detailed evaluation and rationale for my recommendation.

1. Originality:

In recent years, there increasing number of deep learning-based studies to automate terminus extraction. Compared with the previous study, especially with the author's previous paper Loebel et al. (2022), what is the improvement of this study regarding the methodology?

See *Reply of RC2*.

The improvements to *Loebel et al. (2022)* have been emphasized more thoroughly throughout Section 2.

1. Impact:

The main objective of automating the terminus extraction is to produce as many termini as possible. However, compared with the CALFIN (Cheng et al, 2021), which produces 22 678 calving front lines across 66 Greenlandic glaciers from 1972 to 2019, this study seems not improve the temporal resolution, temporal coverage, and spatial coverage. A comparison between the product from this study and CALFIN would be helpful. For instance, which glaciers CALFIN did not cover but this study covers.

The results of this comparison, not only to CALFIN but also to AutoTerm and Termpicks, are presented in the new Section 4.3.

1. Literature Review:

The discussion about the terminus variation and basal topography is interesting, but studies have been investigating this for many years (Joughin et al., 2008, 2014; Kehrl et al., 2017; Bunce et al., 2018, Catania et al., 2018), suggesting that retrograde bed slopes can cause glacier dynamic instabilities (Meier and Post, 1987) and substantial retreat. However, these papers are missing in the manuscript. Therefore, I suggest the author include a literature review about how the glacier geometry influences the terminus variation, also remove the expression throughout this manuscript that this study is the first application analyzing the interaction between calving front variation and bedrock topography.

The discussion (Section 5) has undergone significant reworking. It now includes a comprehensive literature review and a more thorough contextualisation of our results.

Considering the points mentioned above, I believe that rejecting the paper would be appropriate. However, I recommend that the authors be given an opportunity to revise and resubmit their work, addressing the concerns mentioned above. By providing constructive feedback and clear expectations for revision, the authors might have the chance to strengthen their manuscript and overcome the present limitations.

**Specific Comments:**

**Line 51:** Why the usage of multispectral sensor information can increase the temporal solution, compared with using the single band? I was asking because using multi-band information could not yield more images.

See *Reply of RC2.*

Section 2.1 and section 4.3 now also address this point.

**Line 53:** I believe the CALFIN can also resolve sub-seasonal terminus variations as it uses all the available Landsat images.

The phrasing at this and other locations has been revised.

**Line 64:** What modification did the author apply here? Does that modification improve the results? It would be better to have a more detailed description of this.

The descriptions in Section 2.3 have been expanded to include this information.

**Section 2.2:** It would be better if the author can change the name of this section (validation) to avoid confusion. Following the deep learning convention, there are three sets: the training set, the validation set, and the test set. This section is actually about the test set but is called validation.

Also, the descriptions of the validation set should be included in the manuscript.

Done.

**Figure 8:** It would be more helpful to change the second column to the time series of the bed elevation at the terminus. Time series might better reflect the verbal descriptions in the discussion section.

This figure (now Figure 10) has been reworked. It is now much easier to identify the bedrock slope at the calving front for a given date

**Reference**

Loebel, E., Scheinert, M., Horwath, M., Heidler, K., Christmann, J., Phan, L. D., ... & Zhu, X. X. (2022). Extracting Glacier Calving Fronts by Deep Learning: The Benefit of Multispectral, Topographic, and Textural Input Features. *IEEE Transactions on Geoscience and Remote Sensing*, *60*, 1-12.

Cheng, D., Hayes, W., Larour, E., Mohajerani, Y., Wood, M., Velicogna, I., & Rignot, E. (2021). Calving Front Machine (CALFIN): glacial termini dataset and automated deep learning extraction method for Greenland, 1972–2019. The Cryosphere, 15(3), 1663-1675.

Joughin, I., Howat, I., Alley, R. B., Ekstrom, G., Fahnestock, M., Moon, T., Nettles, M., Truffer, M., and Tsai, V. C.: Ice front variation and tidewater behavior on Helheim and Kangerdlugssuaq Glaciers, Greenland, J. Geophys. Res.-Earth, 113, F01004, https://doi.org/10.1029/2007JF000837, 2008.

Kehrl, L. M., Joughin, I., Shean, D. E., Floricioiu, D., and Krieger, L.: Seasonal and interannual variabilities in terminus position, glacier velocity, and surface elevation at Helheim and Kangerlussuaq Glaciers from 2008 to 2016, J. Geophys. Res.-Earth, 122, 1635–1652, https://doi.org/10.1002/2016JF004133, 2017.

Catania, G. A., Stearns, L. A., Sutherland, D. A., Fried, M. J., Bartholomaus, T. C., Morlighem, M., Shroyer, E., and Nash, J.: Geometric Controls on Tidewater Glacier Retreat in Central Western Greenland, J. Geophys. Res.-Earth, 123, 2024–2038, https://doi.org/10.1029/2017JF004499, 2018.

Bunce, C., Carr, J. R., Nienow, P. W., Ross, N., and Killick, R.: Ice front change of marine-terminating outlet glaciers in northwest and southeast Greenland during the 21st century, J. Glaciol., 64, 523–535, https://doi.org/10.1017/jog.2018.44, 2018.

Meier, M. F., & Post, A. (1987). Fast tidewater glaciers. *Journal of Geophysical Research: Solid Earth*, *92*(B9), 9051-9058.

**General comments**

The authors present a deep learning method for automating the digitization of glacier calving fronts from Landsat 8 and 9 images, applied to 23 glaciers in Greenland from 2013 through 2021. They demonstrate the potential application of this method in studying aspects of outlet glacier behavior such as calving front seasonality and interactions with bed topography.

Overall the manuscript is clearly written, and the figures are high quality and do a good job of supporting the text. However, I am concerned that it is lacking important context from previous work, and is not sufficiently distinguishable from other automation methods:

First, how does this deep learning method compare to previous contributions to automation, both in methodology and in outcomes (accuracy)? In other words, as a data user, why should I choose the output from this particular automated method over any of the others that have been recently published? The authors highlight the temporal resolution of this dataset, but to me it doesn't appear to be substantially unique from other automated or even some manual datasets.

The results of this comparison, not only to CALFIN but also to AutoTerm and Termpicks, are presented in the new Section 4.3.

Next, for the highlighted seasonal and topographic applications, how do the results presented here agree with or differ from other studies? There is a wealth of literature on both terminus position seasonality and terminus-topography relationships, but none of it is referenced in the presentation of these data applications, and the authors suggest that their analysis is unprecedented, which is simply not true. Both section 3.2 and the discussion would benefit from a more thorough literature review and contextualization of the work presented.

The discussion (Section 5) has undergone significant reworking. It now includes a comprehensive literature review and a more thorough contextualisation of our results.

Finally, can the authors demonstrate that this automated method, and its resultant higher temporal density of calving fronts, improves upon prior analyses? It would be helpful to move beyond demonstrating the capabilities to demonstrating how they are better than existing approaches. Figure 6 starts to get at this with the comparison between TUD and ESA-CCI, but I think this could be explored in more detail.

This is demonstrated in the comparison presented in the new Section 4.3.

**Specific comments**

L31-35: The authors cite several studies that use manually delineated fronts, then state that these products "often lack temporal resolution, making seasonal analysis … difficult." However, several of the cited products do perform robust seasonal analyses. Schild and Hamilton (2013) had near-daily termini for five glaciers. King et al. (2020) had seasonal data, though centerline positions rather than full calving fronts. Goliber et al. (2022) specifically addressed how their combined dataset could be used to better study seasonality than any of the individual component datasets. Black and Joughin (2023) was explicitly about analyzing sub-seasonal (weekly or monthly) front variability throughout Greenland – in fact, for their weekly dataset, they reported an average of 50 fronts per glacier per year (including polar night), versus 45 in this

manuscript (9243 fronts / 23 glaciers / 9 years). While it is true that manual digitization is laborious and will struggle to keep up with the volume of new imagery (L33), that cannot be used to imply that those efforts have not produced comparable datasets. The authors go on to use language such as "unprecedented temporal resolution, resolving [the glaciers'] sub-seasonal calving front variability for the first time" (L52-53), "[this time series] clearly surpasses the potential of manually delineated data products" (L166-168), and "unprecedented temporal resolution" (L173). Papers cited within this manuscript, as well as others, have repeatedly demonstrated comparable temporal resolution of calving front data products, including through manual digitization.

*The formulations at these and other locations have been reworked. The phrase "unprecedented temporal resolution" has been removed from the manuscript entirely.*

L73-75: I'm glad these various messy situations were considered for training data. Did the training data also focus on glaciers that are heavily crevassed near the terminus? This can also be a tricky condition – and different from dense mélange – perhaps wrapped up in "morphological features"?

*See Reply of RC2.*

*These crevassed conditions are now mentioned in the revised text (section 2.2).*

L78-80: Did this system end up being spatially transferable? The results for looking at the non-Greenlandic glaciers are not addressed.

*Our results on spatial transferability are presented in a new section (section 3.2).*

L98-101: I agree with the overall conclusion that the quality of automated fronts from this method is comparable to the quality of manually delineated termini. The comparison with the error estimate from Goliber et al. (2022) is a helpful reference, but a direct comparison is difficult due to differences in the error calculation method. Goliber et al. (2022) use the Hausdorff distance (they say "greatest minimum distance between two lines") rather than averaging the minimal distances along the front as is done here, so I suspect their error estimate would be a bit larger than what is reported here. It would also be helpful to see how error estimates from this method compare to other automated methods (it looks like this is at least done for CALFIN in Table C1, but this should be mentioned in the main body text about error estimation too).

*The Hausdorff distance estimate is now included in the accuracy assessment in Section 3.1.*

L105-106: What percentage of imagery was completely clouded (and therefore unused)?

*Section 4.1 now includes this information.*

L106-107: What qualifies as a "failed calving front extraction"? What are the criteria for failure? Did the authors manually check all results from the automation?

*Section 4.1 now includes this information.*

Figure 8/Discussion: This is a nice presentation of the relationship between topography and calving fronts over seasons and years. The expanded interpretation in the discussion is helpful, especially for highlighting correlations between topography and seasonal amplitude. However, these interpretations should be placed in the context of other topography-terminus studies (how are they similar/different, what is new here, etc.). Catania et al. (https://doi.org/10.1029/2017JF004499 and others) would be a good starting point.

The discussion (Section 5) has undergone significant reworking. It now includes a comprehensive literature review and a more thorough contextualisation of our results.

Appendix B: Since the core of this manuscript is the deep learning method, the methodology should be in the main body rather than tucked away in an appendix.

We have incorporated appendix A and B into the main body of the manuscript, specifically in Section 2.

**Technical corrections**

L72: change "build" to "built"

Done.

L140: add BedMachine version here (I see v5 in the references, but it is helpful to see up front since there have been some big changes between versions).

Done.

L172: remove "are" after Nioghalvfjerdsbræ

Fixed.

---

## Author Response (AR2)

Dear Bert,

Thank you for your work on our manuscript. We have addressed the remaining points raised by the referees. Please find our revised manuscript together with a track change version. Below, we also provide answers to how and where the referee's comments were addressed.

Best wishes,

Erik and all co-authors

**Anonymous Referee #2, 05 Mar 2024**

Most of the comments have been carefully addressed, and I appreciate the response provided by the author. However, I would like to comment on a limitation regarding the integration of multispectral bands as follows.

The authors demonstrate that integrating multispectral bands can yield more accurate results compared to using single-band inputs, particularly in resolving challenging conditions such as ice-mélange. Nonetheless, there are several limitations to consider. Firstly, the exclusion of the panchromatic band from Landsat-8 impacts the spatial resolution of the terminus products. While I understand that this omission may be necessitated by the requirement of identical resolution for the input bands in the artificial neural network (ANN) used, it raises concerns about potentially sacrificing higher spatial resolution data. Secondly, the requirement of nine bands for input in the ANN restricts the integration of other satellite products. For instance, SAR images like Sentinel-1 typically do not offer multiple bands, and while Sentinel-2 images provide 12 bands, only four of them have the highest 10-meter resolution. Consequently, while the integration of multispectral bands from Landsat-8 provides more termini compared to CALFIN or AutoTerm, it excludes the utilization of other valuable datasets entirely, which in turn affects the temporal resolution of the terminus data. Particularly during winter when optical images lack coverage, SAR images can provide essential winter traces, which are crucial for studying seasonality.

Although I do not currently propose solutions for these limitations, I recommend that the authors at least acknowledge and discuss them in their work.

We thank the reviewer for raising this point, this is indeed worth discussing.

In principle, it is no problem to integrate input data with different resolutions into our processing. In fact, that's what we did with the 100 m resolution TIR bands. This is not a limitation to the use of multispectral information. The reason we chose not to integrate the 15 m panchromatic band is that either the input tiles would have been too large and thus the computational effort too high (if we decided for larger tiles), or we would have lost the spatial context within individual ANN predictions (if we decided for a smaller area).

The nine input bands of our model are tailored to use Landsat-8 and Landsat-9 imagery. Even the integration of Sentinel-2 imagery would require retraining of the model, as Sentinel-2 only captures imagery in VIS and SWIR wavelengths and therefore does not acquire TIR data. Therefore, we fully agree with the reviewers statement that our approach excludes the use of other satellite sensors. In contrast to the approach of *Zhang et. al 2023* (AutoTerm), where the authors developed a model

capable of analyzing multi-sensor imagery, our approach uses only Landsat imagery, but exploits the full sensor information. There are advantages and limitations to both approaches.

We now emphasize these aspects and limitations in several places in our manuscript.

In Section 2.1:

*[...] "The 15 m resolution band 8 is excluded due to the otherwise too high computational cost." [...].*

In Section 2.3.1:

*[...] "The 30 m ground sampling distance, and thus the exclusion of panchromatic band 8, is a compromise between the spatial context provided within a single subset, the computational effort and the resolution of the calving front predictions." [...].*

In Section 4.3:

*[...] "AutoTerm has the most mapped and unique fronts as well as the highest sampling rate. This is mainly due to its data basis which includes Landsat, Sentinel-2 and Sentinel-1. Compared to our approach, which is limited to the use of multispectral Landsat data, this is a clear advantage." [...].*

In Section 4.3:

*[...] "As a final point, we want to emphasise the potential of combining different glacier front products. Particularly for data sets based on optical data, this not only increases the overall sampling rate, but also allows data gaps to be filled during the polar winter." [...].*

**Anonymous Referee #3, 01 Mar 2024**

The authors present a revised manuscript detailing their deep learning method for automated delineation of glacier calving fronts. The method is applied to a set of glaciers in Greenland, but the authors demonstrate spatial transferability to other glaciers in Greenland as well as in Svalbard, Patagonia, and Antarctica. The manuscript discussion has been greatly improved from the previous submission with respect to providing context from previous work on seasonal calving front variations and interactions with bed topography, and I thank the authors for their work to address that and other comments.

I do have some remaining general concerns about the comparisons with other calving front datasets and the evidence provided for distinguishing this method from other automated calving front delineation methods:

The authors have addressed comments concerning the novelty and capabilities of their method by adding a comparison of their model output with that of some other "big data" calving front datasets. The TermPicks dataset (Goliber et al., 2022) is used as a point of reference for comparing this method to manually delineated datasets. This is appropriate as TermPicks is a large compilation of several previously published datasets: it is comparable in size to automatically delineated datasets, it spans the space and time necessary for a comparison with this work, and it provides internal error estimates which can also be compared with this and other methods. However, the analysis presented here (pages 14-15) emphasizes differences in temporal resolution/sampling rate and in the capacity to capture seasonal changes, and the author response (RC-3; not the manuscript) specifically criticizes the

uneven spatial and temporal sampling of TermPicks. While this is a shortcoming of the TermPicks dataset, this is due to it being a compilation of other datasets with disparate spatial and temporal requirements, and not necessarily a weakness of manual delineation more generally as is implied. Black and Joughin (2023), which is already cited elsewhere in the manuscript, is also a "big data" calving front dataset with circum-Greenland coverage, which are the criteria listed for the comparative analysis presented here, and that work is specifically about seasonal changes. Given the emphasis on temporal resolution and seasonal changes presented here, that dataset would be more suitable for the in-depth comparison presented in this manuscript, in addition to the other three datasets already used.

We agree and are thankful for this recommendation. The inclusion of the *Black and Joughin (2023)* data set is a very welcome addition to our comparison.

We included the data set of *Black and Joughin (2023)* into our comparison on section 4.3. Please find the additions in Table 2, Figure 9 and the whole text. We also want to note that we had to change the reference time from 2013-2019 to 2015-2019 which led to changes in the statistics for the other data sets as well.

With regards to evidence for distinguishing this method from other automated methods: the authors highlight this method's success in "challenging conditions" (L276 and in the conclusion). This seems to be the main strength of this method over previous automated delineation methods. However, the only evidence given to support this claim is the specific case of the seasonal ice tongue on Kangiata Nunaata Sermia (L264-269). Can a more general observation of the relative success of this method in challenging conditions be proven? Evidence of this point needs to be strengthened in the results in order to justify the conclusion.

We see why these statements are problematic. Although our method extracted a significant amount of calving front that other methods did not (Section 4.3, Table 2), especially compared to the CALFIN product which is using the same (Landsat) data basis, we agree that this does not necessarily mean that our method is more capable for difficult conditions. However, as our dataset is the only one benefiting from multi-spectral inputs, the integration of which leads to more accurate calving front predictions for challenging ice mélange and illumination conditions (*Loebel et al. 2022*), we believe this is a possible explanation.

The statement in L276 is not substantiated and is changed accordingly. It now reads:

*[...] "Importantly, these 13% are likely to include extractions under challenging ice mélange and illumination conditions." [...].*

The statement in the conclusion is removed entirely.

**Specific comments**

L25-27: Greene (2024) is cited later but would be a good reference to include here as well, to further motivate the importance of calving front retreat for quantifying sea level rise.

Citation is now included.

L32-34: Regarding calving front products lacking temporal resolution: I commented on this paragraph in my previous review, and the authors agreed in the response that it should be changed or clarified, along with several other related comments throughout the manuscript. It seems the other areas were

addressed but this paragraph was perhaps overlooked, as the tracked changes show no edits to this paragraph. To recap, I agree with the statement in the response that manual delineation has not kept up with the volume of satellite imagery, but that is not the sentiment conveyed by this paragraph. Some manual studies have achieved temporal resolution comparable to automated methods, and the comment about the difficulty of performing seasonal analyses is contradicted by multiple papers cited in this paragraph. Please see my previous comment as well and revise.

L53: "the glaciology community requirement" – be specific about what this requirement is that you are meeting.

We have specified this statement. The sentence now reads:

*[...] "By achieving this robust and scalable calving front extraction, we meet the glaciology community requirement for a comprehensive parameterization of glacier calving in Greenland and make important steps towards establishing artificially intelligent processing strategies for glacier monitoring tasks." [...].*

L54 and L346: "intelligent processing strategies" – perhaps say "artificially intelligent" or something similar to highlight the autonomous aspect of this work (previous approaches are not necessarily unintelligent!).

We follow this suggestion and change the phrasing to "artificially intelligent".

L196: When looking at area differences >1km2 between two entries, is this area change scaled by the length of the calving front at all? An area change of 1km2 at Humboldt is relatively small compared to an area change of 1km2 at Hayes, for example.

The area difference is not scaled by the length of the calving front. The intentionally low value of 1 km² was found to be a good compromise for calving fronts captured in a single ANN input tile. For larger glaciers that are delineated by merging several tiles, namely Humboldt Glacier, Zachariæ Isstrøm and Nioghalvfjerdsbrae, this low value resulted in an increased amount of separated extractions resulting in more time spend manually checking these separated extractions. The 1 km² threshold did not negatively affect the quality of the data product or the results of this study.

However, we are very thankful for this suggestion. A scaled threshold, maybe even by calving front length as well as ice velocity, is indeed a good way to further improve our processing in the future. Especially when integrating more large calving fronts, for example in Antarctica.

L281-282: I agree that combining calving front products is helpful to increase spatiotemporal coverage; Goliber et al. (2022) has been cited several times in this manuscript and should be included here, as that is a key reference for combining different calving front products for further analyses.

Citation is now included.

L288: What qualifies a glacier as having "clear seasonality"? How did you determine that 19 glaciers are clearly seasonal? In the results (L218-219) it says that only two glaciers do not have clear seasonality, which would suggest that 21 of the 23 do have clear seasonality, rather than 19.

The "clear seasonality" referred to in this statement has been determined by a visual inspection of the time series. We appreciate this question and we also think that a "clear" seasonality needs a more quantifiable definition. We changed the phrasing of the sentence (L288) to:

*[...] "A visual inspection of the time series shows that 19 of the 23 glaciers analyzed in this study show a seasonal pattern between the years 2013 and 2021." [...].*

Figures of all time series can be found in the supplement.

The statement in L216-219 describes the time series in Figure 8, where there are two glaciers (out of 12) without seasonal calving front variation. In our entire dataset, 19 out of 23 glaciers show seasonal calving front variation.

**Technical corrections**

Generally: correct the inconsistent usage of é in mélange

Done

L20: "…imbalance is driven by changes…" --> "…imbalance is driven in part by changes…" (because the ice-ocean boundary is not the only control on dynamics)

The wording in this paragraph was somewhat vague and sub-optimal, so we have changed it. It now reads:

*[...] "While about half of the ice mass loss is due to increased meltwater runoff, the other half is due to changes of ice discharge to the ocean related to changes of the ice flow dynamics of outlet glaciers (Otosaka et al., 2023). Several mechanisms act as controls and indicators for dynamic glacier changes. In particular, calving and calving front variations have been identified as crucial parameters for investigating the physical mechanisms of Greenland outlet glaciers (Joughin et al., 2008a; Moon and Joughin, 2008; Benn et al., 2017; Trevers et al., 2019; Cook et al., 2021; Melton et al., 2022)." [...]*

Figure 5c caption: Patagonia

Fixed

L193: "Depending on" (not "of")

Done

L338: Humboldt

Fixed

**References**

Black, T. E. and Joughin, I.: Weekly to monthly terminus variability of Greenland's marine-terminating outlet glaciers, The Cryosphere, 17, 1–13, https://doi.org/10.5194/tc-17-1-2023, 2023.

Goliber, S., Black, T., Catania, G., Lea, J. M., Olsen, H., Cheng, D., Bevan, S., Bjørk, A., Bunce, C., Brough, S., Carr, J. R., Cowton, T., Gardner, A., Fahrner, D., Hill, E., Joughin, I., Korsgaard, N. J., Luckman, A., Moon, T., Murray, T., Sole, A., Wood, M., and Zhang, E.: TermPicks: a century of Greenland glacier terminus data for use in scientific and machine learning applications, The Cryosphere, 16, 3215–3233, https://doi.org/10.5194/tc-16-3215-2022, 2022.

Greene, C. A., Gardner, A. S., Wood, M., and Cuzzone, J. K.: Ubiquitous acceleration in Greenland Ice Sheet calving from 1985 to 2022, Nature, 625, 523–528, https://doi.org/10.1038/s41586-023-06863-2, 2024.

Loebel, E., Scheinert, M., Horwath, M., Heidler, K., Christmann, J., Phan, L. D., ... & Zhu, X. X. (2022). Extracting Glacier Calving Fronts by Deep Learning: The Benefit of Multispectral, Topographic, and Textural Input Features. *IEEE Transactions on Geoscience and Remote Sensing*, *60*, 1-12.

---

## Author Response (AR3)

Dear Bert,

Thank you very much for the work you did on our manuscript. We have corrected the errors listed below.

Best wishes,

Erik and all co-authors

L36-37: R3 raised a point about this paragraph, but it seems it was not addressed. Please adjust this sentence to address R3's comment. For example, "Thus, such calving front products may not always exploit the full temporal information offered by the satellite observations, which may be a limiting factor in seasonal analysis and associated modelling efforts."

You are right, we apologize for this oversight. We have changed the sentence according to your suggestion. Thank you very much.

L259: "glaciers by considering the temporal (2013 to 2019)" – shouldn't this be 2015-2019? Please check this throughout the manuscript.

We have corrected this and checked the entire section. Thank you.

L270: "A clear advantage over our approach, which is limited to the use of multispectral Landsat data." – This is not a proper sentence, a verb is missing in the main sentence, please adjust (e.g. 'This is a clear…")

Following your suggestion, we have corrected this sentence.